# Preparation and Characterization of Natural Silk Fibroin Hydrogel for Protein Drug Delivery

**DOI:** 10.3390/molecules27113418

**Published:** 2022-05-25

**Authors:** Junwei Liu, Haowen Sun, Yuwei Peng, Ligen Chen, Wei Xu, Rong Shao

**Affiliations:** 1School of Chemistry and Chemical Engineering, Yancheng Institute of Technology, Yancheng 224051, China; liujunwei_1222@163.com; 2School of Marine and Bioengineering, Yancheng Institute of Technology, Yancheng 224051, China; shw987052988@163.com (H.S.); pengyvwei1114@163.com (Y.P.); ycit549638894@outlook.com (L.C.); xuweiyc@163.com (W.X.); 3Jiangsu Key Laboratory of Biochemistry and Biotechnology of Marine Wetland, Yancheng Institute of Technology, Yancheng 224051, China

**Keywords:** silk fibroin, hydrogel, recycled material, drug delivery

## Abstract

In recent years, hydrogels have been widely used as drug carriers, especially in the area of protein delivery. The natural silk fibroin produced from cocoons of the Bombyx mori silkworm possesses excellent biocompatibility, significant bioactivity, and biodegradability. Therefore, silk fibroin-based hydrogels are arousing widespread interest in biomedical research. In this study, a process for extracting natural silk fibroin from raw silk textile yarns was established, and three aqueous solutions of silk fibroin with different molecular weight distributions were successfully prepared by controlling the degumming time. Silk fibroin was dispersed in the aqueous solution as “spherical” aggregate particles, and the smaller particles continuously accumulated into large particles. Finally, a silk fibroin hydrogel network was formed. A rheological analysis showed that as the concentration of the silk fibroin hydrogel increased its storage modulus increased significantly. The degradation behavior of silk fibroin hydrogel in different media verified its excellent stability, and the prepared silk fibroin hydrogel had good biocompatibility and an excellent drug-loading capacity. After the protein model drug BSA was loaded, the cumulative drug release within 12 h reached 80%. We hope that these investigations will promote the potential utilities of silk fibroin hydrogels in clinical medicine.

## 1. Introduction

With the development of science and technology, problems related to resources and the environment have become prominent. In the process of the industrial production of silk fabrics, tons of damaged or difficult-to-handle silkworm cocoons are discarded as waste every year due to their substandard quality [1]. In addition, waste textile products in daily life are everywhere and are increasing day by day. This has caused great wastage of resources and, to a certain extent, serious environmental problems [2,3]. The use of silk dates back thousands of years. It is rich in output, has excellent mechanical properties, and is known as the “Queen of Fiber” for its luster color sense and soft, delicate feel. It has been widely used in textiles and clothing [3,4,5]. With the gradual development and application of the efficacy of fibroin in silk, silk has been favored in biomedicine as a valuable natural protein fiber raw material [6]. Like natural silk fibers, the main components of cocoon waste and waste fabrics are silk fibroin and sericin, which can be used as raw materials for the development of silk-based products [7]. Therefore, recycled silk waste still has a considerable recycling value left to be explored and broad application prospects in the field of medical hydrogels (Figure 1A). The benefits are significant in terms of alleviating the storage pressure of waste silk and reducing the cost of silk products.

Taking “carbon neutrality” as an opportunity, using low-cost natural resources to produce valuable functional products will improve resource utilization [8], promote energy conservation and emission reduction, build a green and low-carbon circular development system, and create a “Green Factory”. At present, among the many developed silk-based biomaterials, silk fibroin hydrogels derived from silkworm cocoons have received considerable attention due to their excellent biological properties, internal extracellular-matrix-like structure, and diverse gelation methods [9,10,11,12]. However, there are few reports on the extraction and properties of silk fibroin from silk waste as a raw material.

As a typical representative of natural materials, silk fibroin can be used in the construction of microcarrier drug-delivery systems in order to effectively control the release behavior of drugs and improve their therapeutic efficiency [13]. Silk fibroin can self-assemble into microcarrier structures such as hydrogels, microspheres, nanoparticles, and hollow microcapsules. Among these, silk fibroin hydrogels have the advantages of good biocompatibility, high drug-encapsulation efficiency, a stable spatial network structure, and a controllable drug-release rate. It has shown good application prospects in drug delivery related to small molecules, proteins/peptides, and nucleic acids [14]. In recent years, research on silk fibroin as a controlled drug-release carrier has been gradually emerging. Compared with commonly used polymer materials, such as poly(lactic-co-glycolic acid) (PLGA), silk fibroin may have broader application prospects in the development of delivery systems for protein polypeptide drugs, nucleic acids, and other biological macromolecular drugs [15].

As a natural product, silk fibroin has the advantages of abundant sources, easy production, good biocompatibility, low immunogenicity, and high safety [16,17]. Its application has a long history. For example, it has been used as a surgical suture for hundreds of years due to its excellent properties. The degummed silk obtained after boiling and degumming is silk fibroin. SEM observation shows that silk fibroin is fibrous and insoluble in water with high degrees of orientation and crystallinity, a compact structure, and superior stability [18]. Silk fibroin hydrogel is an important macroscopic form of protein materials. It has flexibility and plasticity and is also permeable to gases and some small molecular substances. It is used for the preparation of cell culture scaffolds, wearable sensors, flexible electronic skins, enzyme immobilization, drug-release carriers, and other biomedical materials [19,20,21,22]. However, the morphology of silk fibroin in aqueous solution and its properties lack attention.

It is of great significance to develop silk fibroin hydrogels and to broaden their biomedical applications. In this work, raw silk textile yarns were used as a raw material to prepare silk fibroin aqueous solutions with three molecular weights by controlling the degumming time. The micromorphological changes in silk fibroin during the sol-gel transformation were observed by SEM. The cytotoxicity of silk fibroin hydrogels was detected by an MTT assay. In addition, the stability of silk fibroin hydrogels was evaluated by degradation behavior studies in different media. Finally, bovine serum albumin (BSA) was used as a drug model to evaluate silk fibroin hydrogel’s protein drug-release properties. This provides a promising strategy for using recycled silk waste instead of silk cocoons from natural sources to prepare silk-based hydrogels as protein/peptide carriers.

## 2. Results and Discussion

### 2.1. Preparation and Characterization of Silk Fibroin

Degummed silk with different molecular weights can be obtained by heating and boiling for different times (0.5, 1, and 2 h) under the same power. The silk fibroin extracted under degumming times of 0.5, 1, and 2 h were named RSF-0.5h, RSF-1h, and RSF-2h, respectively. The prepared degummed silk was dissolved in 9.3 mol/L lithium bromide (LiBr) solution to obtain silk fibroin aqueous solutions with three molecular weights. The preparation process of silk fibroin aqueous solution is shown in Figure 1B [23].

An inductively coupled plasma emission spectrometry analysis showed that the Li^+^ contents in the silk fibroin aqueous solutions with three molecular weights were all at very low concentrations (*p* < 0.01), and the quality was acceptable (Appendix A). According to the Malvern laser particle size analysis, the relative molecular weights of the silk fibroin obtained after degumming times of 0.5, 1, and 2 h were 1590 ± 244, 1280 ± 143, and 917 ± 38 kDa, respectively. Under the same concentration, gel times of RSF-0.5h, RSF-1h, and RSF-2h were prolonged successively, indicating that the decrease in molecular weight can slow down the gelation rate of silk fibroin solution (Figure 1). The overall hydrodynamic diameters of RSF-0.5h, RSF-1h, and RSF-2h were 651.7 ± 4.7, 152.7 ± 0.4, and 73.8 ± 0.5 nm, respectively, indicating that silk fibroin with a large molecular weight corresponds to a large particle size (Appendix A). When effectively removing sericin, the molecular weight of silk fibroin decreases with the prolongation of degumming time, which may be caused by a partial degradation of the silk fibroin [24].

The UV spectra showed that the aqueous solutions of silk fibroin with different molecular weights did not differ significantly at the molecular level, as shown in Appendix A. An obvious ultraviolet characteristic absorption peak was observed at 275.5 nm, which was due to the presence of abundant aromatic amino acids in the molecular chain of silk fibroin [25]. The freshly prepared silk fibroin aqueous solutions with three molecular weights and their hydrogels were freeze-dried, and the elemental composition of the lyophilized samples was determined by an elemental analyzer. The C/N and C/H atomic ratios of the aqueous solution samples were 2.6 and 7, respectively, the C/N and C/H atomic ratios of the hydrogel samples were 2.6 and 7.2, respectively, and the C/N atomic ratios of the two were equal (Appendix A). The C/H atomic ratio of the hydrogel sample was slightly higher than that of the aqueous solution sample, which was caused by the difference in the secondary structure of silk fibroin [26]. The aggregated structure of silk fibroin was analyzed by FT-IR and XRD, and it was found that the aqueous solution samples were mainly composed of a silk I (α-helix and random coil) structure, while the hydrogel samples were mainly composed of a silk II (β-sheet) structure, and the crystallinity of the hydrogel samples was higher (Appendix A). The thermal stability analysis of the samples using TGA showed that the thermal degradation process of silk fibroin was mainly concentrated in the range of 280–400 °C, with excellent thermal stability (Appendix A). In addition, there was an exothermic peak on the DSC curve of the aqueous solution sample, while this was absent on the DSC curve of the hydrogel sample. This further verifies the conclusions of FT-IR and XRD.

The silk fibroin sol-gel transformation process is affected by concentration, pH, and temperature. In this work, the effect of these parameters on gelation rate was explored. The results showed that with increasing concentrations of silk fibroin solution, pH value, and temperature, the gelation rate of the silk fibroin solution accelerated (Table 1). Considering the homeostasis of the organism’s internal environment and the practical application of silk fibroin hydrogels, the silk fibroin hydrogels for in vitro degradation and drug-release behavior studies were prepared with initial concentrations of silk fibroin aqueous solution of 30 and 20 mg/mL at 37 °C and pH 7.4 (Figure 1C).

### 2.2. Micromorphology Analysis of Silk Fibroin Sol-Gel Transformations

The sol-gel transformation process of silk fibroin aqueous solution (RSF-1h) with concentrations of 15 mg/mL (RSF15) and 10 mg/mL (RSF10) was observed by SEM, and the results are shown in Figure 2. The silk fibroin was dispersed in the aqueous solution in the form of “spherical” aggregate particles. With the extension of time, the silk fibroin in the aqueous solution spontaneously aggregated to form larger “spherical” particles. The measured particle sizes of RSF15 at 0, 12, and 24 h were 47.41 ± 1.72, 107.48 ± 2.89, and 155.48 ± 3.91 nm, and the particle sizes of RSF10 at 0, 12, and 24 h were 38.07 ± 4.17, 85.41± 4.78, and 145.56 ± 2.18 nm, respectively. Obviously, the particle size increased with the extension of time. Before gelation, the “pellets” were continuously clustered together into “big balls” and then accumulated with each other to form aggregates. The aggregates were interconnected to form a porous network with a certain spatial conformation, thereby forming a hydrogel.

### 2.3. Mechanical Properties of Silk Fibroin Hydrogels

The storage modulus, G′, and the loss modulus, G″, are also known as the elastic modulus and the viscous modulus, which reflect the elasticity and the viscosity of the system, respectively. When the sizes of G′ and G″ are close, the system is semi-solid. The rheological properties of the hydrogels after the gelation of silk fibroin aqueous solutions with different concentrations (30, 20, 15, 10, and 5 mg/mL) are shown in Figure 3.

Figure 3A shows the changes in the storage modulus, G′, and loss modulus, G″, of silk fibroin hydrogel RSF-1h at five concentrations within 600 s; G′ and G″ had no cross-linking point and satisfy G′ > G″, indicating that the silk fibroin aqueous solutions with concentrations of 30, 20, 15, 10, and 5 mg/mL gelled, and G′ was much larger than G″. The material mainly assumed an elastic deformation and eventually presented a solid state. The difference between G′ and G″ increased with the increase in silk fibroin concentration over time within 600 s, indicating that the silk fibroin hydrogel was a good elastomer with a certain mechanical strength [27]. In addition, from Figure 3A, it can be observed that the difference, ΔG, between G′ and G″ of the five concentrations of RSF-1h silk fibroin samples was sorted as follows: 30 mg/mL > 20 mg/mL > 15 mg/mL > 10 mg/mL > 5 mg/mL, and the loss moduli were approximately equal, indicating that the hydrogels formed with high silk fibroin contents had better mechanical properties. Among them, the storage modulus of the hydrogel formed by 30 mg/mL silk fibroin aqueous solution reached 13.4 kPa.

Figure 3B shows the changes in the storage modulus, G′, and loss modulus, G″, of silk fibroin hydrogel RSF-0.5h at five concentrations within 600 s. The rheological properties of RSF-0.5h with the change in the concentration of silk fibroin aqueous solution was consistent with that of RSF-1h; that is, the greater the concentration of silk fibroin aqueous solution, the greater the storage modulus of the hydrogel formed by it, and the storage modulus reflected the elastic capacity, indicating that the high-concentration silk fibroin hydrogel had a higher mechanical strength [28]. The storage modulus of the hydrogel formed by 30 mg/mL silk fibroin aqueous solution reached 19.0 kPa. In addition, comparing the rheological properties of RSF-0.5h and RSF-1h, it was found that the silk fibroin with a larger molecular weight had a higher mechanical strength after forming a hydrogel at the same concentration.

### 2.4. Cytotoxicity Evaluation of Silk Fibroin Hydrogels

The cytotoxicity of silk fibroin hydrogels was investigated using HepG2 cells. After co-culturing the leaching solutions of three silk fibroin hydrogel samples (24, 48, and 72 h) with HepG2 cells for 48 h, the relative cell viability of HepG2 cells was calculated using the MTT method. The experimental results are shown in Figure 4. Compared with the blank control group, after co-culture with cells for 48 h, the relative viability of HepG2 cells remained at around 100% in the leaching solutions obtained under different leaching times (24, 48, and 72 h) of silk fibroin hydrogels with three molecular weights, indicating good cell growth. In particular, the relative viability of cells after co-incubating with the 72 h leaching solution for 48 h was still not significantly lower than 100%; that is, there was no obvious cell death phenomenon. The above analysis indicated that the silk fibroin hydrogels had no significant cytotoxicity [5,29]. Therefore, the prepared silk fibroin hydrogel had good biocompatibility and could be used in studies on drug release.

Meanwhile, the cell morphology of the experimental group and the control group was photographed with an inverted microscope to assist in verifying the results of the MTT analysis. The photographing situation is shown in Figure 5. After 48 h of co-incubation, the experimental group was compared with the control group. It was found that the cells grew well as a whole, and a large number of living cells proliferated and diffused successfully, presenting a consistent spindle shape, which further confirmed the detection results of the MTT method. These results indicated that the silk fibroin hydrogel had excellent biocompatibility and was suitable for controlled drug release or as a wound dressing carrier in biomedicine [30,31,32].

### 2.5. Degradation Behavior Evaluation of Silk Fibroin Hydrogels

The degradation behavior of hydrogels formed from silk fibroin (RSF-1h) obtained by degumming for 1 h was investigated in this work. Figure 6A,B show that the silk fibroin hydrogel had no obvious degradation in PB buffers with different pH values but displayed a slight swelling phenomenon. The silk fibroin hydrogel had a three-dimensional porous network structure with a large specific surface area [33], which can absorb and lock part of the water; in addition, some salt ions entered the hydrogel, resulting in dense pores, limiting the water absorption capacity of the hydrogel and reaching a swelling balance [24].

The degradation behavior of silk fibroin hydrogel in acid and alkali conditions was remarkable. With the increase in the concentration of hydrochloric acid, the degradation rate of RSF15 increased, especially in a 2 mol/L hydrochloric acid solution. The degradation rate increased significantly after the 10th day, and the mass decreased almost exponentially; it was completely degraded on the 21st day, while the degradation was slower at other concentrations (Figure 6C). However, the degradation rate of RSF10 was faster (Figure 6D) because the β-sheet structure of RSF10 was reduced and the strength was lower, leading to accelerated degradation [34]. RSF10 was completely degraded at 11, 16, and 18 days in 2, 1, and 0.5 mol/L hydrochloric acid solution, respectively, and the mass decreased by 75.03% at 22 days in 0.1 mol/L hydrochloric acid solution. It can be seen from Figure 6E,F that when the sodium hydroxide solution was used as a degradation medium, the higher the concentration, the more obvious the degradation degree, and the silk fibroin hydrogel could be completely degraded in the solution in 1–2 days. It is worth pointing out that the sodium hydroxide solution was different from the other degradation media; it directly “melted” the hydrogel into a paste-like viscous liquid within a certain period of time.

When an organic solvent was used as the degradation medium, the degradation behavior of the silk fibroin hydrogel was related to the polarity of the organic solvent (the relationship of polarity was: DMSO > DMF > ethanol). Solvent molecules with strong polarity can easily form hydrogen bonds with the hydrophilic groups of hydrogels; thus, the solvent molecules can quickly penetrate the hydrogel network. DMF and anhydrous ethanol with poor polarity had a poor ability to form hydrogen bonds and had a certain water absorption capacity [9,35]. Therefore, as shown in Figure 6G,H, the hydrogel exhibited a swelling phenomenon in DMSO, slight dehydration in DMF, and obvious dehydration in anhydrous ethanol.

With PBS buffer (pH 7.4) as a blank control, elastase and reduced glutathione were dissolved in PBS buffer as a degradation medium. Figure 6I,J show that the silk fibroin hydrogels displayed no obvious degradation in the first two weeks and slightly swelled. In the third week, there was a slight degradation. Silk fibroin hydrogels had slow in vitro degradation performance in elastase and reduced glutathione solutions, among which the degradation behavior of hydrogels with high silk fibroin content was slower at the same ionic strength. This may be related to the higher content of β-sheet conformation in silk fibroin hydrogels [36].

As shown in Figure 6K,L, the silk fibroin hydrogel displayed no degradation in a NaCl solution and maintained a slight swelling state within 22 days. The higher the salt concentration, the more obvious the swelling phenomenon of the hydrogel, which may be due to the entry of salt ions after swelling, forming a more stable crystal structure inside the hydrogel. This limited the degradation behavior to a certain extent [37]. The higher the salt concentration, the stronger this limiting effect. In addition, it was observed that when the silk fibroin hydrogel RSF10 was in deionized water, the surface of the pore wall collapsed and fragmented, and the integrity of the overall structure of the hydrogel was damaged.

### 2.6. Drug Loading and Release Properties of Silk Fibroin Hydrogels

The photographs of RSF10 FITC-BSA-containing hydrogels with different drug loading contents (DLCs) are shown in Appendix A. By increasing the FITC-BSA loading content up to 30%, the hydrogels presented an undesirable dehydration phenomenon. While the DLC of RSF10 was kept to 20% or below, an intact FITC-BSA-loaded silk fibroin hydrogel could be formed. Therefore, the maximum DLC of RSF10 to form an intact hydrogel was 20%, and all fed FITC-BSA drug was encapsulated into the hydrogel in this process with a drug loading efficiency of 100%. Besides, the photographs of RSF15 FITC-BSA-containing hydrogels with different DLCs are shown in Appendix A. According to the observations, the maximum DLC of RSF15 to form an intact hydrogel without dehydration was determined to be 70%. The drug loading efficiency in this hydrogel was also 100%.

To avoid the waste of silk fibroin, a high DLC to form an intact drug-containing hydrogel is beneficial. However, it was observed that a higher DLC led to a higher gel-strength but also a dehydration phenomenon, as shown in Appendix A. In order to obtain appropriate FITC-BSA-loaded silk fibroin hydrogels with moderate DLCs and adequate gel strengths as the candidates for the evaluation of drug release profiles, the final DLCs of BSA-loaded hydrogel in this experiment were set at 20% and 40%. In detail, the silk fibroin solution with an initial concentration of 30 mg/mL and the FITC-BSA solution were selected for equal volume mixing to prepare FITC-BSA-loading hydrogels with the DLCs of 40% and 20% for drug release experiments. After drug encapsulation, the hydrogels with DLCs of 40% and 20% were tagged as DLC40 and DLC20, respectively and the final RSF concentrations in the two hydrogels were 15 mg/mL. Appendix A shows the variation in the storage modulus, G′, and the loss modulus, G″, of DLC40 and DLC20 after gelation. The two modulus curves had no cross-linking point, and G′ was greater than G″, indicating that the elastic properties of the hydrogel were dominant, and DLC40 exhibited a higher strength.

The cumulative drug-release curves of the drug-containing hydrogels with different drug loadings are shown in Figure 7. It can be seen in the figure that the overall drug-release trends of DLC40 and DLC20 were the same, and the cumulative drug-release rate of DLC40 was higher than that of DLC20 at the same point in time. This is because DLC40 encapsulated a higher concentration of BSA, which led to more diffusion of BSA molecules into the release solution. The drug-release behavior was divided into three stages, namely, a 2–12 h burst-release stage, a 12–72 h sustained-release stage, and a 72–168 h full-release stage. When the drug was released for 12 h, the cumulative drug release of DLC40 and DLC20 reached 80.08 ± 1.22% and 73.82 ± 2.82%, respectively. The release curves at this stage were approximately linear, and the release rate was faster, which may be due to the rapid release of some drugs at the junction of the gel and the release solution into the PBS buffer during the initial swelling of the hydrogel [38]. With the prolongation of the release time, the slope of the tangent point of the release curve gradually became smaller; that is, the release rate became slower. When the drug was released for 72 h, the cumulative drug release of DLC40 and DLC20 reached 98.37 ± 1.62% and 91.24 ± 3.26%, respectively, which may be due to the continuous occurrence of swelling behavior, such that the BSA encapsulated in the hydrogel was released. The concentration of BSA inside the hydrogel gradually decreased, resulting in slower molecular diffusion, which manifested slower drug release [39]. In addition, the degradation of the hydrogel and the interaction between the drug and the internal hydrogel structure may lead to a decrease in drug release. When the drug-release time exceeded 72 h, the drug-release trends of DLC40 and DLC20 were flat and slightly declined. This may be related to the internal β-sheet structure of silk fibroin hydrogels, meaning that the remaining drugs were encapsulated within them and were not easy to release [31]. The cumulative drug release values at 168 h were 97.89 ± 2.08% and 91.44 ± 3.02%, respectively, indicating that the drug release had reached saturation and the drug could be considered to be no longer being released or finished in a state of equilibrium. The results show that the silk fibroin hydrogel had a good release effect on BSA. The drug release curve shows that the drug-containing hydrogel could be released within 12 h. The cumulative drug release could reach 80%, indicating that the silk fibroin hydrogel has a potential application as a protein drug carrier.

## 3. Materials and Methods

### 3.1. Materials

Raw silk textile yarns from silkworm cocoons were purchased from Zhejiang Haiyan Jinyi Silk Spinning Co., Ltd., Jiaxing, China; sodium carbonate, lithium bromide, sodium chloride, glutathione (GSH), N,N-dimethylformamide (DMF), and dimethyl sulfoxide (DMSO) were purchased from Aladdin Chemical Reagent Co., Ltd., Shanghai, China; phosphate-buffered saline (PBS) was purchased from Hangzhou Baisi Biotechnology Co., Ltd., Hangzhou, China; anhydrous ethanol, hydrochloric acid, and sodium hydroxide were purchased from Sinopharm Chemical Reagent Co., Ltd., Shanghai, China; 3-(4,5-dimethyl-2-thiazolyl)-2,5-diphenyltetrazolium bromide (MTT) was purchased from Sigma-Aldrich (Shanghai, China); elastase (from porcine pancreas, 30 U/mg) was purchased from Beijing Jinming Biotechnology Co., Ltd., Beijing, China; fluorescein isothiocyanate isomer I (FITC) was purchased from Innochem (Beijing, China); bovine serum albumin (BSA) and dialysis bags (MWCO 3500 Da) were purchased from Shanghai Yuanye Bio-Technology Co., Ltd., Shanghai, China; and liquid nitrogen was purchased from Yancheng Guangyuan Gas Co., Ltd., Yancheng, China. The chemical reagents were all analytical grade (AR), and the experimental water was ultrapure water/deionized water.

### 3.2. Testing and Characterization

The samples were treated by wet nitrification, and the lithium element in the sample solution was determined using an inductively coupled plasma emission spectrometer (ICP, OPTIMA 8000DV, PerkinElmer, Waltham, MA, USA). The silk fibroin aqueous solution was serially diluted with ultrapure water to a concentration range of 0.1–1.0 mg/mL. Based on the static light scattering (SLS) theory, the test temperature of the instrument was set to 25 °C, a Malvern laser particle size analyzer (Zetasizer Nano ZS, Malvern, UK) was used to measure the light-scattering intensity of the sample solution to be tested relative to the standard with a known Rayleigh ratio, and the molecular weight of silk fibroin was calculated by Rayleigh Equation (1). Based on the dynamic light scattering (DLS) theory, the particle size distribution of the sample solution to be tested was measured by a Malvern laser particle size analyzer [40]. In addition, the gel time of the silk fibroin aqueous solution was observed and recorded at an ambient temperature of 4 °C.
*KC*/*R*_*θ*_ = 1/*M* + 2*A*_2_*C*(1)
where *C* is the concentration, *M* is the molecular weight of the sample, *A*_2_ is the second virial coefficient, *R**_θ_* is the Rayleigh ratio, the ratio of the scattered light of the sample to the incident light, and *K* is the optical constant.

The samples were tested for UV absorption using a full-wavelength UV spectrometer (UV-2450, Shimadzu, Japan), and the wavelength scanning range was set to 200–400 nm. The contents of C, N, and H in the samples were determined by an elemental analyzer (Vario EL Cube, Elementar, Hanau, Germany). A certain mass of freeze-dried samples was weighed and placed on the test bench of a Fourier-transform infrared spectrometer (FT-IR, NEXUS-670, Nicolet, Madison, WI, USA) for spectrum collection. The scanning range was 4000–400 cm^−1^, the number of scans was 64, and the resolution was 4 cm^−1^. The internal crystal structure of the samples was tested using an X-ray diffractometer (XRD, X’Pert3 Powder, PANalytical, The Netherlands), where the diffraction angle range was 5–80°, the scanning speed was 2°/min, the current was 40 mA, the tube voltage was 40 kV, and CuKα rays were used. Under the protection of high-purity N_2_, the temperature was increased from room temperature to 800 °C at a heating rate of 10 °C/min, and the thermal stability of the samples was analyzed and determined using a TG-DSC synchronous thermal analyzer (STA 449C, NETZSCH, Selb, Germany).

### 3.3. Experimental Methods

#### 3.3.1. Micromorphology Analysis

The silk fibroin aqueous solution was mixed with PBS buffer at a volume ratio of 1:1 and shaken, then placed in a 37 °C incubator to incubate until a hydrogel was formed. During this period, a small amount of solution was dropped on the activated silicon wafer every 12 h and left to stand overnight in a dry and clean area. After the water was fully evaporated and dried, the silicon wafer containing the sample was pasted on the electron microscope sample stage with a conductive adhesive, and gold was sprayed with a spray current of 30 mA in a vacuum state for 2 min [41]. The microscopic morphology of the sample surface was observed under a scanning electron microscope (Nova NanoSEM 450, Hillsboro, OR, USA) with a shooting voltage of 15 kV.

#### 3.3.2. Rheological Analysis

The silk fibroin aqueous solution was diluted with ultrapure water to 30, 20, 15, 10, and 5 mg/mL, and incubated in a 37 °C incubator to form a hydrogel. The G′ and G″ of the hydrogel samples with different contents of silk fibroin were tested in the “Time Mode” using a rheometer (DHR-3, New Castle County, DE, USA) at a constant frequency of 10 rad/s, and the diameter of the test parallel plate was 40 mm [42]. The gap was set to 1 mm, and the temperature was set to 37 °C. In order to avoid the loss of water from the hydrogel, the samples were tested after being quickly placed on the experimental bench of the instrument.

#### 3.3.3. Cytotoxicity Assay

The cytotoxicity of the silk fibroin hydrogels was evaluated by measuring the survival rate of human hepatoma cells (HepG2 cells) after co-culture in a silk fibroin hydrogel leaching solution for a period of time by an MTT assay. Under sterile conditions, we retrieved the 96-well plate in which the cells were grown adherently and added 100 μL of silk fibroin hydrogel sample leaching solution and 100 μL of fresh DMEM medium containing 10% fetal bovine serum and 1% double antibody (100 U/mL penicillin and 100 μg/mL streptomycin) to each well and cultured the plate in a cell incubator at 37 °C with 5% CO_2_ [43]. After 48 h of incubation in the incubator, we added 20 μL of MTT solution (1 mg/mL) to each well. After a further 4 h of incubation, we carefully discarded the supernatant and added 150 μL of DMSO solution to each well, shook for 20 min, and measured the absorbance of each well with a microplate reader at a wavelength of 490 nm. In addition, the cell morphology (bright field) of the silk fibroin hydrogel sample leaching solution in the experimental group after co-culture with HepG2 cells for 48 h was photographed with an inverted microscope, and the morphology of the cells was compared with the control group.

We calculated the relative cell viability according to Equation (2):*Relative cell viability* (%) = *A*_*s*_/*A*_0_(2)
where *A_s_* is the absorbance value of the experimental group and *A*_0_ is the absorbance value of the control group.

In this work, an experimental group and a blank control group were set up. The experimental group contained the 24, 48, and 72 h leaching solutions of silk fibroin hydrogels (RSF-0.5h, RSF-1h, and RSF-2h), and the leaching solutions were diluted 10 times. The blank control group was the medium containing 10% fetal bovine serum and 1% double antibody (100 U/mL penicillin and 100 μg/mL streptomycin). The experimental group and blank control group were added to the 96-well plate inoculated with cells, and six parallel samples were set in each group. Finally, the 96-well plate was co-cultured in a cell incubator at 37 °C and 5% CO_2_ for 48 h.

#### 3.3.4. Degradation in Different Media

The degradation behavior of two concentrations of silk fibroin hydrogels (RSF15 and RSF10) in different media (phosphate buffers with different pH values, hydrochloric acid, and sodium hydroxide solutions with different concentrations, organic solvents, protease solutions, and NaCl solutions with different concentrations) were explored.

We prepared a number of dry and clean glass vials, numbered and weighed them, formed hydrogels in the glass vials, and weighed the total mass of the glass vials and the hydrogel after gelation. The fresh degradation solution was added to the glass vial containing the hydrogel at a ratio of 1:1 (*v*/*v*), and the degradation experiment was carried out in an incubator at 37 °C [44]. The samples were replaced with fresh degradation solution every day. The samples were taken out according to the set time, the degradation solution was discarded, and the total mass of the glass vial and the remaining hydrogel was weighed. The residual mass-retention rate of the hydrogel can be calculated according to Equation (3).
*Remaining mass retention rate* (%) = (*M*_*i*_ − *M*)/(*M*_0_ − *M*) × 100(3)
where *M*_0_ is the initial total mass of the glass vial and hydrogel, g; *M* is the mass of the glass vial, g; and *M_i_* is the total mass of the glass vial and remaining hydrogel after *i* days, g.

#### 3.3.5. Drug Loading and Release

The FITC-labeled BSA was used as a protein drug model, which was dissolved in PBS buffer and mixed by vortexing to obtain a drug solution with a specific concentration [45]. According to Formula (4), the drug loading content (*DLC*) was set to 10, 20, 30, 40, 50, 60, 70, 80, and 90%, and the concentration of the protein drug solution was calculated. The silk fibroin aqueous solution and the protein drug solution were mixed in equal volumes to obtain drug-containing silk fibroin solutions with final concentrations of 15 and 10 mg/mL, respectively. The drug-containing hydrogels were obtained by incubation in a 37 °C incubator under dark conditions to promote drug encapsulation.

*DLC* represents the percentage of the total mass of the drug in the drug-containing hydrogel, and the specific calculation formula is as follows in Equation (4):*DLC* (%) = *c*_*d*_*V*_*d*_/(*c*_*d*_*V*_*d*_ + *c*_*h*_*V*_*h*_) × 100(4)
where *c_d_* is the concentration of the drug solution, mg/mL; *V_d_* is the volume of the drug solution, mL; *c_h_* is the concentration of the pre-drug-loaded silk fibroin aqueous solution, mg/mL; and *V_h_* is the volume of the pre-drug-loaded silk fibroin aqueous solution, mL.

The PBS buffer was used as the drug-release solution, and each drug-containing hydrogel was tested in parallel in three groups. An equal volume of PBS buffer was added to the drug-containing hydrogel glass vial, and the drug was released in a 37 °C constant temperature incubator at a speed of 80 r/min. At the set time points (2, 4, 6, 8, 12, 24, 36, 48, 60, 72, 96, 120, 144, and 168 h), we drew an appropriate amount of drug-release solution and immediately injected the same amount of fresh PBS buffer, keeping the volume of the release solution constant and continuing to release the drug in the incubator. We diluted this sample of release solution 20 times with fresh PBS buffer, then diluted it in half and stepwise until it became a colorless and transparent liquid. We recorded the final dilution factor.

The fluorescence spectrum of the diluted drug-release solution was collected using a JASCO FP-6500 fluorescence spectrometer, and the amount of the drug released at the corresponding time point was calculated according to the standard curve of fluorescently labeled BSA. Then, the cumulative drug-release curve of the drug-containing hydrogel was drawn.

The final drug quality data calculated from the standard curve (Appendix A) were the mean values of the three groups of parallel experiments, and the cumulative drug release rate of the drug-containing hydrogel was calculated according to Equation (5):*Cumulative release rate* (%) = *M*_*t*_/*M*_*d*_ × 100(5)
where *M_t_* is the mass of the drug released at time *t*, mg, and *M_d_* is the initial mass of the loaded drug, mg.

## 4. Conclusions

In conclusion, a complete and stable silk fibroin extraction and aqueous solution preparation process was established using raw silk textile yarns as stock materials. Silk fibroins with different molecular weights can be obtained by controlling the degumming time. The silk fibroin was dispersed in the aqueous solution as “spherical” aggregate particles, and smaller particles continuously accumulated into larger particles. Finally, a silk fibroin hydrogel network was formed. The secondary structures of silk fibroin aqueous solution and hydrogel lyophilized samples were different. The former was dominated by a silk I structure, while the latter was dominated by a silk II structure, and the crystallinity of the hydrogel samples was higher. When the silk fibroin hydrogel concentration increased, its storage modulus increased significantly. By studying the degradation behavior of silk fibroin hydrogel in various media, it was verified that it had excellent stability, which can further broaden its potential applications. Moreover, the prepared silk fibroin hydrogel had good biocompatibility and excellent drug loading capacity, and the cumulative drug release reached 80% within 12 h after loading BSA, indicating the potential utility of this hydrogel for the delivery of protein drugs.

## Data Availability

The data generated or analyzed during this study are available from the corresponding author on reasonable request.

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
