# Peer review of "Preparation and Characterization of Natural Silk Fibroin Hydrogel for Protein Drug Delivery"

_molecules, 2022, doi:10.3390/molecules27113418_

Round 1

Reviewer 1 Report

The manuscript “Preparation and Characterization of Natural Silk Fibroin Hydrogel for Protein Drug Delivery” reports new experimental data on the obtaining of silk fibroin hydrogel loaded with model protein drug BSA.

The manuscript must be subjected to a major revision in order to be considered for publication. See below the main observations.

1) The introduction is too long, I suggest to be reduced by removing the general aspects (for example about the use of hydrogels) and focus on the aim of the study.

2) The paragraph starting with line 97 describes the methods used for preparation and characterization of the silk hydrogel, all these details must be moved to the Material and Methods section.

3) In the Introduction section authors must highlight the aim of their study.

4) The sentence at line 141 must be explained, supported by the data.

5) Line 149: “It was preliminarily determined by UV analysis that the chemical composition of silk fibroin in aqueous solution was not affected by molecular weight” UV spectra is not an elective method for determination of chemical composition. How the authors expect that molecular weight will influence the chemical composition of the silk fibroin?

This paragraph must be clarified.

6) Line 161: “The thermal stability analysis of the samples by TGA and DSC shows that the thermal degradation residue of the hydrogel samples is slightly higher than that of the aqueous solution samples.” The authors must provide an explanation.

7) Notes at Table 1 is very confusing, the authors must rephrase.

8) Figure 4 must be split, the images in Figure 4B (cell morphology) must be at higher size to allow reader to observe differences.

9) An English language revision must be performed and style improved (See paragraph starting line 244, that must be rewritten).

10) The same observation for paragraph at line 258, is very confusing.

11) The paragraph at line 262 about swelling does not have connection with degradation, I suggest to be moved to another section, maybe morphology.

12) “2.6. Drug Release of Drug-Loaded Hydrogels” The release is of the drug, not of the hydrogel. The title of the section must be changed “Drug release” (it is obvious that the release is from the drug loaded hydrogel).

13) The sentence at line 358 “Within a certain drug loading range, the larger the drug loading, the better the drug release effect” does not have any sense. What the authors mean with “drug release effect”? The phrase must be rewritten or removed.

14) Some comments on the specificity of the hydrogels (not only the code) in relationship with the drug release profile and with the degradation will be beneficial, in order to emphasize the effect of the morphology or other physic-chemical properties of the hydrogels on those important characteristics.

15) Line 147 “Compared with external drugs, the process of internal drugs from inside to the outside takes a certain amount of time, which is manifested as slower drug release .” What the authors intent to say with “internal drugs” ?

16) Line 443 “and then the quality of the drug released”? Probably it is the quantity (amount) of drug released?

17) Line 454 “preparation process was formed” A process could not be formed, the phrase must be rewritten

18) It is not clear which is a novelty of the research. The relevance of the work must be evidenced in the Conclusions section.

Author Response

Replies to your comments:

Firstly, we would like to express our great thanks for your constructive suggestions on our manuscript. After going through our paper carefully, we found that all the comments are helpful for us to revise our manuscript. Here, all scientific questions arisen from your comments are answered in detail one by one, hopefully the overall quality of this manuscript could reach the criteria for publishing in Molecules.

Response to the reviewer 1:

The manuscript “Preparation and Characterization of Natural Silk Fibroin Hydrogel for Protein Drug Delivery” reports new experimental data on the obtaining of silk fibroin hydrogel loaded with model protein drug BSA.

The manuscript must be subjected to a major revision in order to be considered for publication. See below the main observations.

Comment 1: The introduction is too long, I suggest to be reduced by removing the general aspects (for example about the use of hydrogels) and focus on the aim of the study.

Our Reply: Thank you for your constructive suggestion. According to your comments, we have rewritten the introduction section in the revised manuscript and it’s presented below. In this section, many redundant and basic descriptions have been removed or concentrated. We have also paid more attention on the main theme of this manuscript. We have proposed and emphasized the aims of this work in the revised introduction section. Silk fibroin is mainly derived from silkworm cocoons in the published works. To the best of our knowledge, studies on the extraction of silk fibroin from raw silk textile yarns to prepare hydrogels as protein drug carriers are rare. We developed a method to use silk fibroin hydrogel derived from raw silk textile yarns as drug carrier in this manuscript, and hope that it could be a potential utility for the recycled silk waste to be used in biomedical area. On the other hand, the degradation experiments of silk fibroin hydrogel were evaluated in a various of media, which presented its interesting degradation behaviors under different conditions. Besides, we thought it would be more appropriate to place Scheme 1 in the corresponding section of "2. Results and Discussion" after careful consideration. Therefore, we moved Scheme 1 to the "2.1. Preparation and Characterization of Silk Fibroin" section.

Introduction Section of the revised manuscript:

With the development of science and technology, problems related to resources and the environment have become prominent. In the process of industrial production of silk fabrics, tons of damaged or difficult-to-handle silkworm cocoons are discarded as waste every year due to their substandard quality [1]. In addition, waste textile products in daily life are everywhere, and are increasing day by day. This has caused great wastage of resources, and to a certain extent, serious environmental problems [2,3]. The use of silk dates back thousands of years. It is rich in output, has excellent mechanical properties, and is known as the “Queen of Fiber” for its luster color sense and soft, delicate feel. It has been widely used in textiles and clothing [3-5]. With the gradual development and application of the efficacy of fibroin in silk, silk has been favored in biomedicine as a valuable, natural, protein fiber raw material [6]. Like natural silk fibers, the main components of cocoon waste and waste fabrics are silk fibroin and sericin, which can be used as raw materials for the development of silk-based products [7]. Therefore, recycled silk waste still has a considerable recycling value left to be explored, and broad application prospects in the field of medical hydrogels (Scheme 1A). The benefits are significant in terms of alleviating the storage pressure of waste silk and reducing the cost of silk products.

Taking “carbon neutrality” as an opportunity, using low-cost natural resources to produce valuable functional products will improve resource utilization [8], promote energy conservation and emission reduction, build a green and low-carbon circular development system, and create a “Green Factory”. At present, among the many developed silk-based biomaterials, silk fibroin hydrogels derived from silkworm cocoons have received considerable attention due to their excellent biological properties, internal extracellular matrix-like structure, and diverse gelation methods [9-12]. However, there are few reports on the extraction and properties of silk fibroin from silk waste as raw material.

As a typical representative of natural materials, silk fibroin can be used in the construction of microcarrier drug-delivery systems in order to effectively control the release behavior of drugs and improve their therapeutic efficiency [13]. Silk fibroin can self-assemble into microcarrier structures such as hydrogels, microspheres, nanoparticles, and hollow microcapsules. Among these, silk fibroin hydrogels have the advantages of good biocompatibility, high drug-encapsulation efficiency, stable spatial network structure and controllable drug-release rate. It has shown good application prospects in drug delivery related to small molecules, proteins/peptides, and nucleic acids [14]. In recent years, research on silk fibroin as a controlled drug-release carrier has been gradually emerging. Compared with commonly used polymer materials such as poly(lactic-co-glycolic acid) (PLGA), silk fibroin may have broader application prospects in the development of delivery systems for protein polypeptide drugs, nucleic acids and other biological macromolecular drugs [15].

As a natural product, silk fibroin has the advantages of abundant sources, easy production, good biocompatibility, low immunogenicity and high safety [16,17]. Its application has a long history. For example, it has been used as a surgical suture for hundreds of years due to its excellent properties. The degummed silk obtained after boiling and degumming is silk fibroin. SEM observation shows that silk fibroin is fibrous and insoluble in water with high degrees of orientation and crystallinity, a compact structure, and superior stability [18]. Silk fibroin hydrogel is an important macroscopic form of protein materials. It has flexibility and plasticity, and is also permeable to gases and some small molecular substances. It is used for the preparation of cell culture scaffolds, wearable sensors, flexible electronic skins, enzyme immobilization, drug-release carriers and other biomedical materials [19-22]. However, the morphology of silk fibroin in aqueous solution and its properties lack attention.

It is of great significance to develop silk fibroin hydrogels and to broaden their biomedical applications. In this work, raw silk textile yarns were used as raw material to prepare silk fibroin aqueous solution with three molecular weights by controlling the degumming time. The micromorphological changes in silk fibroin during sol–gel transformation were observed by SEM. The cytotoxicity of silk fibroin hydrogels was detected by MTT assay. In addition, the stability of silk fibroin hydrogels was evaluated by degradation behavior studies in different media. Finally, bovine serum albumin (BSA) was used as a drug model to evaluate silk fibroin hydrogel’s protein drug-release properties. This provides a promising strategy for using recycled silk waste instead of silk cocoons from natural sources to prepare silk-based hydrogels as protein/peptide carriers.

References

  1. Lamoolphak, W.; De-Eknamkul, W.; Shotipruk, A. Hydrothermal production and characterization of protein and amino acids from silk waste. Bioresource Technol. 2008, 99, 7678-7685.
  2. Liu, H.; Wei, J.; Zheng, L.J.; Zhao, Y.P. Extraction and characterization of silk fibroin from waste silk. Mater. Res. 2013, 788, 174-177.
  3. Zamani, B.; Svanström, M.; Peters, G.; Rydberg, T. A carbon footprint of textile recycling: a case study in sweden. Ind. Ecol. 2015, 19, 676-687.
  4. Tian, Y.; Jiang, X.J.; Chen, X.; Shao, Z.Z.; Yang, W.L. Doxorubicin-loaded magnetic silk fibroin nanoparticles for targeted therapy of multidrug-resistant cancer. Mater. 2014, 26, 7393-7398.
  5. Luo, K.Y.; Yang, Y.H.; Shao, Z.Z. Physically crosslinked biocompatible silk-fibroin-based hydrogels with high mechanical performance. Funct. Mater. 2016, 26, 872-880.
  6. Elliott, W.H.; Bonani, W.; Maniglio, D.; Motta, A.; Tan, W.; Migliaresi, C. Silk hydrogels of tunable structure and viscoelastic properties using different chronological orders of genipin and physical cross-linking. ACS Appl. Mater. Inter. 2015, 7, 12099-12108.
  7. Ghalei, S.; Handa, H. A review on antibacterial silk fibroin-based biomaterials: current state and prospects. Today Chem. 2022, 23, 100673.
  8. Nogueira, G.M.; Rodas, A.C.D.; Leite, C.A.P.; Giles, C.; Higa, O.Z.; Polakiewicz, B.; Beppu, M.M. Preparation and characterization of ethanol-treated silk fibroin dense membranes for biomaterials application using waste silk fibers as raw material. Bioresource Technol. 2010, 101, 8446-8451.
  9. Zheng, H.Y.; Zuo, B.Q. Functional silk fibroin hydrogels: preparation, properties and applications. Mater. Chem. B 2021, 9, 1238-1258.
  10. Jiang, M.R.; Shu, T.; Ye, C.; Ren, J.; Ling, S.J. Predicting the conformations of the silk protein through deep learning. Analyst 2021, 146, 2490-2498.
  11. Dorishetty, P.; Dutta, N.K.; Choudhury, N.R. Silk fibroins in multiscale dimensions for diverse applications. RSC Adv. 2020, 10, 33227-33247.
  12. Zhang, X.; Jiang, S.T.; Yan, T.F.; Fan, X.T.; Li, F.; Yang, X.D.; Ren, B.; Xu, J.Y.; Liu, J.Q. Injectable and fast self-healing protein hydrogels. Soft Matter 2019, 15, 7583-7589.
  13. Tomeh, M.A.; Hadianamrei, R.; Zhao, X.B. Silk fibroin as a functional biomaterial for drug and gene delivery. Pharmaceutics 2019, 11, 494.
  14. Yucel, T.; Lovett, M.L.; Kaplan, D.L. Silk-based biomaterials for sustained drug delivery. Control. Release 2014, 190, 381-397.
  15. Hofmann, S.; Stok, K.S.; Kohler, T.; Meinel, A.J.; Müller, R. Effect of sterilization on structural and material properties of 3-D silk fibroin scaffolds. Acta Biomater. 2014, 10, 308-317.
  16. Vepari, C.; Kaplan, D.L. Silk as a biomaterial. Polym. Sci. 2007, 32, 991-1007.
  17. Tan, H.P.; Marra, K.G. Injectable, biodegradable hydrogels for tissue engineering applications. Materials 2010, 3, 1746-1767.
  18. Kim, H.J.; Um, I.C. Effect of degumming ratio on wet spinning and post drawing performance of regenerated silk. J. Biol. Macromol. 2014, 67, 387-393.
  19. Chen, Z.Y.; Wang, Y.; Zhao, Y.J. Bioinspired conductive cellulose liquid-crystal hydrogels as multifunctional electrical skins. Natl. Acad. Sci. USA 2020, 117, 18310-18316.
  20. Wang, C.Y.; Xia, K.L.; Wang, H.M, Liang, X.P.; Yin, Z.; Zhang, Y.Y. Advanced carbon for flexible and wearable electronics. Mater. 2019, 31, 1801072.
  21. Ye, J.J.; Chu, T.S.; Chu, J.L.; Gao, B.B.; He, B.F. A versatile approach for enzyme immobilization using chemically modified 3D-printed scaffolds. ACS Sustain. Chem. Eng. 2019, 7, 18048-18054.
  22. Song, W.T.; Das, M.; Xu, Y.D.; Si, X.H.; Zhang, Y.; Tang, Z.H.; Chen, X.S. Leveraging biomaterials for cancer immunotherapy: targeting pattern recognition receptors. Today Nano 2019, 5, 100029.

Comment 2: The paragraph starting with line 97 describes the methods used for preparation and characterization of the silk hydrogel, all these details must be moved to the Material and Methods section.

Our Reply: Thank you a lot. We removed the methods used for preparation and characterization of the silk hydrogels in the Introduction section of the original manuscript. All these details are supplemented in the Materials and Methods section of the revised manuscript and labeled in red. The detailed descriptions are as follows.

Introduction Section of the original manuscript

It is of great significance to develop silk fibroin hydrogels to broaden their biomedical applications. In this paper, raw silk textile yarns were used as raw material to prepare silk fibroin aqueous solution with three molecular weights by controlling the degumming time. The molecular weight and particle size distribution of silk fibroins were determined by Malvern laser particle size analyzer. The silk fibroin aqueous solution and its hydrogel samples were freeze-dried, and the freeze-dried samples were characterized by elemental analysis, infrared spectroscopy, X-ray diffraction, thermogravimetric analysis and differential scanning calorimetry, and their chemical composition, aggregated structure and thermal stability were analyzed. The rheological analysis of silk fibroin hydrogel was carried out using a rheometer to determine its mechanical strength. The micromorphological changes of silk fibroin during the sol-gel transformation were observed by SEM, and the formation mechanism of silk fibroin hydrogels was preliminarily explored.

Testing and Characterization Section of the revised manuscript

3.2. Testing and Characterization

The samples were treated by wet nitrification, and the lithium element in the sample solution was determined using an inductively coupled plasma emission spectrometer (ICP, OPTIMA 8000DV, PerkinElmer, USA). The silk fibroin aqueous solution was serially diluted with ultrapure water to a concentration range of 0.1-1.0 mg/mL. Based on the static light scattering (SLS) theory, the test temperature of the instrument was set to 25 °C, and the Malvern laser particle size analyzer (Zetasizer Nano ZS, Malvern, UK) was used to measure the light-scattering intensity of the sample solution to be tested relative to the standard with known Rayleigh ratio, and the molecular weight of silk fibroin was calculated by Rayleigh equation (1). Based on the dynamic light scattering (DLS) theory, the particle size distribution of the sample solution to be tested was measured by a Malvern laser particle size analyzer [41]. In addition, the gel time of the silk fibroin aqueous solution was observed and recorded at an ambient temperature of 4 °C.

KC/Rq = 1/M + 2A2C                        (1)

where C is the concentration, M is the molecular weight of the sample, A2 is the second virial coefficient, Rq is the Rayleigh ratio—the ratio of the scattered light of the sample to the incident light, and K is the optical constant.

The samples were tested for UV absorption using a full-wavelength UV spectrometer (UV-2450, Shimadzu, Japan), and the wavelength scanning range was set to 200-400 nm. The contents of C, N, and H in the samples were determined by elemental analyzer (Vario EL Cube, Elementar, Germany). A certain mass of freeze-dried samples was weighed and placed on the test bench of a Fourier-transform infrared spectrometer (FT-IR, NEXUS-670, Nicolet, USA) for spectrum collection. The scanning range was 4000-400 cm-1, the number of scans was 64, and the resolution was 4 cm-1. The internal crystal structure of the samples was tested using an X-ray diffractometer (XRD, X’Pert3 Powder, PANalytical, Netherlands), where the diffraction angle range was 5-80°, the scanning speed was 2°/min, the current was 40 mA, the tube voltage was 40 kV, and CuKα rays were used. Under the protection of high-purity N2, the temperature was increased from room temperature to 800 °C at a heating rate of 10 °C/min, and the thermal stability of the samples was analyzed and determined using a TG-DSC synchronous thermal analyzer (STA 449C, NE-TZSCH, Germany).

References

  1.  
  2.  
  3.  
  4.  
  5.  
  6.  
  7.  
  8.  
  9.  
  10.  
  11.  
  12.  
  13.  
  14.  
  15.  
  16.  
  17.  
  18.  
  19.  
  20.  
  21.  
  22.  
  23.  
  24.  
  25.  
  26.  
  27.  
  28.  
  29.  
  30.  
  31.  
  32.  
  33.  
  34.  
  35.  
  36.  
  37.  
  38.  
  39.  
  40.  
  41.  
  42.  
  43.  
  44.  
  45.  
  46.  
  47.  
  48.  
  49.  
  50.  
  51.  
  52.  
  53.  
  54.  
  55. Liu, J.W.; Ding, Z.Z.; Lu, G.Z.; Wang, J.G.; Wang, L.; Lu, Q. Amorphous silk fibroin nanofiber hydrogels with enhanced mechanical properties. Biosci. 2019, 19, 1900326.

Comment 3: In the Introduction section authors must highlight the aim of their study.

Our Reply: Thanks for your suggestion. We have rewritten the introduction section in the revised manuscript. In this section, many redundant descriptions have been removed or concentrated. We have paid more attention on the main theme of this manuscript. The aim of our study is highlighted below.

We have proposed and emphasized the aims of this work in the revised introduction section. Silk fibroin is mainly derived from silkworm cocoons in the published works. To the best of our knowledge, studies on the extraction of silk fibroin from raw silk textile yarns to prepare hydrogels as protein drug carriers are rare. We developed a method to use silk fibroin hydrogel derived from raw silk textile yarns as drug carrier in this manuscript, and hope that it could be a potential utility for the recycled silk waste to be used in biomedical area. In addition, the gelation properties of silk fibroin hydrogels under different conditions were investigated in detail. We found that the concentrations of RSF, the environmental temperatures and the pH values would all influence the gelation behaviors of silk fibroin. On the other hand, the degradation experiments of silk fibroin hydrogel were evaluated in a various of media, which presented its interesting degradation behaviors under different conditions.

Comment 4: The sentence at line 141 must be explained, supported by the data.

Our Reply: Thanks a lot for your comments. The description “In addition, too short degumming time may lead to incomplete degumming, too long degumming time may lead to the decomposition of some silk fibroin into smaller molecular fragments.” in Sub-section 2.1 of the original manuscript has been removed. The reason is that this statement is conjectural opinion, and we have no corresponding experimental data to support this statement. Therefore, it is removed in the revised manuscript.

Comment 5: Line 149: “It was preliminarily determined by UV analysis that the chemical composition of silk fibroin in aqueous solution was not affected by molecular weight” UV spectra is not an elective method for determination of chemical composition. How the authors expect that molecular weight will influence the chemical composition of the silk fibroin? This paragraph must be clarified.

Our Reply: Thanks for your constructive suggestions. We strongly agree with the reviewer that UV spectra is indeed not an elective method for determination of chemical composition. Therefore, we consulted and referred to the explanation in the latest literature "Bio-derived crystalline silk nanosheets for versatile macroscopic assemblies (DOI: 10.1007/s12274-022-4124-x)". The description “It was preliminarily determined by UV analysis that the chemical composition of silk fibroin in aqueous solution was not affected by molecular weight.” in Sub-section 2.1 of the original manuscript has been redescribed. The redescription in this section of the revised manuscript is labeled in red and it’s presented below.

The UV spectra showed that the aqueous solutions of silk fibroin with different molecular weights did not differ significantly at the molecular level, as shown in Figure S3. An obvious ultraviolet characteristic absorption peak was observed at 275.5 nm, which was due to the presence of abundant aromatic amino acids in the molecular chain of silk fibroin [25]. The freshly prepared silk fibroin aqueous solutions with three molecular weights and their hydrogels were freeze-dried, and the elemental composition of the lyophilized samples was determined by an elemental analyzer. The C/N and C/H atomic ratios of the aqueous solution samples are 2.6 and 7, respectively, the C/N and C/H atomic ratios of the hydrogel samples are 2.6 and 7.2, respectively, and the C/N atomic ratios of the two are equal (Table S1). The C/H atomic ratio of the hydrogel sample is slightly higher than that of the aqueous solution sample, which is caused by the difference in the secondary structure of silk fibroin [26].

Figure S3. UV spectra of silk fibroin aqueous solutions with different molecular weights. (A) RSF-0.5h; (B) RSF-1h; (C) RSF-2h.

References

  1.  
  2.  
  3.  
  4.  
  5.  
  6.  
  7.  
  8.  
  9.  
  10.  
  11.  
  12.  
  13.  
  14.  
  15.  
  16.  
  17.  
  18.  
  19.  
  20.  
  21.  
  22.  
  23.  
  24.  
  25. Cheng, B.C.; Lei, Z.Y.; Wu, P.Y. Bio-derived crystalline silk nanosheets for versatile macroscopic assemblies. Nano Res. 2022, 10.1007/s12274-022-4124-x.
  26. Nagarkar, S.; Patil, A.; Lele, A.; Bhat, S.; Bellare, J.; Mashelkar, R.A. Some mechanistic insights into the gelation of regenerated silk fibroin sol. Ind. Eng. Chem. Res. 2009, 48, 8014-8023.

Comment 6: Line 161: “The thermal stability analysis of the samples by TGA and DSC shows that the thermal degradation residue of the hydrogel samples is slightly higher than that of the aqueous solution samples.” The authors must provide an explanation.

Our Reply: Thanks a lot for your comments. According to your comment, we realize that this statement is inappropriate. Figure S5 shows that the profiles of thermal degradation residues of the two groups are very close. Therefore, we have removed the statement “The thermal stability analysis of the samples by TGA and DSC shows that the thermal degradation residue of the hydrogel samples is slightly higher than that of the aqueous solution samples.” in Sub-section 2.1 of the original manuscript.

Figure S5. Thermal stability analysis of silk fibroin. TGA (A) and DSC (B) diagrams of silk fibro-in aqueous solutions with different molecular weights. TGA (C) and DSC (D) diagrams of silk fibroin hydrogels with different molecular weights.

Comment 7: Notes at Table 1 is very confusing, the authors must rephrase.

Our Reply: Thanks for your suggestion. We have rephrased the Notes of the revised manuscript and labeled in red.

"+" indicates that the silk fibroin solution gels within 24 h, "+ +" indicates that the silk fibroin solution gels within 48 h, "+ + +" indicates that the silk fibroin solution gels within 72 h, "+ + + +" indicates that the silk fibroin solution gels within 96 h. "-" indicates that the silk fibroin solution did not gel within 10 days.

Comment 8: Figure 4 must be split, the images in Figure 4B (cell morphology) must be at higher size to allow reader to observe differences.

Our Reply: Thanks for your advice. We have split Figure 4 into two figures (Figure 4 and Figure 5), so that Figure 5 can have a larger size for the readers to observe the differences. Figures 4 and 5 have been reset in the revised manuscript as follows.

2.4. Cytotoxicity Evaluation of Silk Fibroin Hydrogels

Cytotoxicity of silk fibroin hydrogels was investigated using HepG2 cells. After co-culturing the leaching solutions of three silk fibroin hydrogel samples (24, 48 and 72 h) with HepG2 cells for 48 h, the relative cell viability of HepG2 cells was calculated using the MTT method. The experimental results are shown in Figure 4. Compared with the blank control group, after co-culture with cells for 48 h, the relative viability of HepG2 cells remained at around 100% in the leaching solutions obtained under different leaching times (24, 48 and 72 h) of silk fibroin hydrogels with three molecular weights, indicating good cell growth. In particular, the relative viability of cells after co-incubating with the 72 h leaching solution for 48 h was still not significantly lower than 100%, that is, there was no obvious cell death phenomenon. The above analysis indicated that the silk fibroin hydrogels had no significant cytotoxicity [29,30]. Therefore, the prepared silk fibroin hydro-gel has good biocompatibility and can be used in studies on drug release.

Figure 4. Effects of silk fibroin hydrogel extracts on the viability of HepG2 cells.

Meanwhile, the cell morphology of the experimental group and the blank control group was photographed with an inverted microscope to assist in verifying the results of the MTT analysis. The photographing situation is shown in Figure 5. After 48 hours of co-incubation, the experimental group was compared with the blank. It was found that the cells grew well as a whole, and a large number of living cells proliferated and diffused successfully, presenting a consistent spindle shape, which further confirmed the detection results of the MTT method. These results indicate that the silk fibroin hydrogel has excel-lent biocompatibility and is suitable for controlled drug release or as a wound dressing carrier in biomedicine [31-33].

Figure 5. Cell morphology observed under an inverted microscope after co-culture of silk fibroin hydrogel extract with HepG2 cells for 48 h. Scale bar is 100 mm.

Comment 9: An English language revision must be performed and style improved (See paragraph starting line 244, that must be rewritten).

Our Reply: Thanks for your suggestion. We regret there were problems with the English languages. We have carefully checked and revised this manuscript. Furthermore, this manuscript has undergone a professional language editing service (MDPI), and the text has been carefully revised to improve grammar errors and readability. Therefore, the readers may understand our work more clearly. The language polishing certificate is as follows. Finally, thanks again for your careful work on our manuscript.

Comment 10: The same observation for paragraph at line 258, is very confusing.

Our Reply: Thanks for your suggestion. In our current work, we only found that the silk fibroin hydrogel RSF10 was degraded in PB buffer at pH 6.0, but the reason for this phenomenon cannot be explained clearly. In view of this phenomenon, it may be related to the isoelectric point of amino acids, resulting in partial acidolysis of protein in PB buffer at pH 6.0, which promotes the degradation of low-content silk fibroin hydrogel (RSF10). In this paper, due to the limitations of the work, we can only describe the observed phenomena and give possible reasons in our opinion. Obviously, the explanation we gave is far from enough, which requires us to further study the degradation properties of silk fibroin hydrogels in acidic PB buffer, and we will give more detailed explanations in future work. Therefore, the experimental data related to pH 6.0 has been removed in the revised manuscript.

Comment 11: The paragraph at line 262 about swelling does not have connection with degradation, I suggest to be moved to another section, maybe morphology.

Our Reply: Thanks for your advice. The samples observed by SEM were dry, while the research object of this part was hydrogels containing a large amount of water, and there is little correlation between the two. Besides, hydrogels often show a swollen state first during the degradation process, so this part of the content is appropriate here.

Comment 12: “2.6. Drug Release of Drug-Loaded Hydrogels” The release is of the drug, not of the hydrogel. The title of the section must be changed “Drug release” (it is obvious that the release is from the drug loaded hydrogel).

Our Reply: Thanks for your suggestion. Your proposal was quite specific and detailed, and we have carefully revised the manuscript. We modified the title of Section 2.6. to "Drug Loading and Release Properties of Silk Fibroin Hydrogels" in the revised manuscript and labeled in red.

Comment 13: The sentence at line 358 “Within a certain drug loading range, the larger the drug loading, the better the drug release effect” does not have any sense. What the authors mean with “drug release effect”? The phrase must be rewritten or removed.

Our Reply: Thanks for your suggestion. The statement “Within a certain drug loading range, the larger the drug loading, the better the drug release effect” in Sub-section 2.6 of the original manuscript has been removed. For "drug release effect", what we want to express is that a large drug load can release more drug to satisfy our expectations. But now, with your reminder, we believe that this statement is superfluous and does not have any sense. Therefore, we have removed this statement in the revised manuscript after comprehensive consideration.

Comment 14: Some comments on the specificity of the hydrogels (not only the code) in relationship with the drug release profile and with the degradation will be beneficial, in order to emphasize the effect of the morphology or other physic-chemical properties of the hydrogels on those important characteristics.

Our Reply: Thanks for your suggestion. In our work, SEM confirmed that the hydrogel was a porous network, and in vitro degradation experiments verified that the hydrogel remained stable after immersion in PBS for 22 days. Thus, the BSA-release behavior could be attributed to the diffusion of BSA molecules due to the concentration gradient difference. Indeed, silk fibroin molecules would also diffuse into the external release-medium from the gel-networks in the drug-release experiments. However, the amount of dissociated silk fibroin was rather low.

Comment 15: Line 147 “Compared with external drugs, the process of internal drugs from inside to the outside takes a certain amount of time, which is manifested as slower drug release.” What the authors intent to say with “internal drugs”?

Our Reply: Thanks for your constructive suggestion. The statement “Compared with external drugs, the process of internal drugs from the inside to the outside takes a certain amount of time, which is manifested as slower drug release.” in Sub-section 2.6 of the original manuscript is not understandable and has been removed. With regards to this, we rephrased this sentence as “The concentration of BSA inside the hydrogel gradually decreased, resulting in slower molecular diffusion, which manifests slower drug release.” in Sub-section 2.6 of the revised manuscript.

Comment 16: Line 443 “and then the quality of the drug released”? Probably it is the quantity (amount) of drug released?

Our Reply: Thank you a lot. It is really inappropriate to use "quality" in this sentence, we have revised it to "amount" in the revised manuscript. We labeled it in red in the revised manuscript and pasted it below.

3.3.5. Drug Release

The fluorescence spectrum of the diluted drug release solution was collected by JASCO FP-6500 fluorescence spectrometer, and then the amount of the drug released at the corresponding time point was calculated according to the standard curve of fluorescently labeled BSA, and then the cumulative drug release curve of the drug-loaded hydrogel was drawn.

Comment 17: Line 454 “preparation process was formed” A process could not be formed, the phrase must be rewritten.

Our Reply: Thanks for your comments. We changed "formed" to "established" in the problematic sentence. The revised word is labeled in red, and presented below:

Original manuscript:

In conclusion, a complete and stable silk fibroin extraction and aqueous solution preparation process was formed by using raw silk textile yarns silk as stock materials in this work.

Revised manuscript:

In conclusion, a complete and stable silk fibroin extraction and aqueous solution preparation process was established by using raw silk textile yarns silk as stock materials in this work.

Comment 18: It is not clear which is a novelty of the research. The relevance of the work must be evidenced in the Conclusions section.

Our Reply: Thanks for your comments. The previously published research articles about this area are insightful and informative. Compared with these published literatures, the main features and novelty of our work are elaborated below.

  • In the published work, silk fibroin is mainly derived from silkworm cocoons, and there are relatively few studies on the extraction of silk fibroin from raw silk textile yarns. At the same time, our work has studied in detail the molecular weight distribution and micromorphology of silk fibroin in aqueous solution by regulating the degumming time of raw silk textile yarns, which has not been reported in previous published studies.

  • The effects of concentration, pH and temperature on the gelation rate of silk fibroin solution were studied in detail, and a silk fibroin hydrogel with moderate molecular weight distribution and concentration, which could gel at physiological pH and temperature, was preferred.

  • We also comprehensively investigated the degradation behavior of silk fibroin hydrogel in different media. Moreover, the silk fibroin hydrogel prepared in this work has an excellent drug loading capacity, whose DLC can reach up to 70% to form an intact hydrogel without dehydration phenomenon.

  • We proposed and emphasized the idea of extracting silk fibroin from silk waste. We hope to develop a method of using silk fibroin hydrogel derived from raw silk textile yarns as protein drug carrier, and also hope that it can provide a preliminary reference for the recycled silk waste to be used in biomedical engineering.

Moreover, according to your comments, we re-summarize the work of the full text and have rewritten the Conclusions section in the revised manuscript. In this section, many redundant descriptions have been removed. The relevance of the work could be evidenced in the following Conclusions section.

Conclusions Section of the revised manuscript

In conclusion, a complete and stable silk fibroin extraction and aqueous solution preparation process was established by using raw silk textile yarns as stock materials. Silk fibroins with different molecular weights can be obtained by controlling the degumming time. The silk fibroin was dispersed in the aqueous solution as "spherical" aggregate particles, and smaller particles continuously accumulated into larger particles. Finally, a silk fibroin hydrogel network was formed. The secondary structures of silk fibroin aqueous solution and hydrogel lyophilized samples are different. The former is dominated by Silk I structure, while the latter is dominated by Silk II structure, and the crystallinity of the hydrogel samples is higher. When the silk fibroin hydrogel concentration increased, its storage modulus increased significantly. By studying the degradation behavior of silk fibroin hydrogel in various media, it has been verified that it has excellent stability, which can further broaden its potential applications. Moreover, the prepared silk fibroin hydrogel has good biocompatibility and excellent drug loading capacity, and the cumulative drug release reaches 80% within 12 h after loading BSA, indicating the potential utility of this hydrogel for the delivery of protein drugs.

Reviewer 2 Report

The current manuscript provides an interesting account of how to employ silk in drug delivery. The method of purification and obtaining silk hydrogel is industrially applicable and provides a very good account of its biomedical application. I have two suggestions to further improve the manuscript:

  1. How stable/intact was the silk fibroin structure after all the processing? No physicochemical characterization is provided.
  2. BSA is a protein and so is silk fibroin. The authors should quantify the protein-protein interactions and how this is affecting or impacting the drug release from the hydrogels.

Author Response

Replies to your comments:

Firstly, we would like to express our great thanks for your constructive suggestions on our manuscript. After going through our paper carefully, we found that all the comments are helpful for us to revise our manuscript. Here, all scientific questions arisen from your comments are answered in detail one by one, hopefully the overall quality of this manuscript could reach the criteria for publishing in Molecules.

Response to the reviewer 2:

The current manuscript provides an interesting account of how to employ silk in drug delivery. The method of purification and obtaining silk hydrogel is industrially applicable and provides a very good account of its biomedical application. I have two suggestions to further improve the manuscript:

Comment 1: How stable/intact was the silk fibroin structure after all the processing? No physicochemical characterization is provided.

Our Reply: Thank you for your suggestion. The corresponding physicochemical characterizations are provided in Supplementary Materials. According to our characterizations, we can know that the silk fibroin structure is well-integrated and can remain relatively stable, which are consistent with the published literatures, such as “Determining beta-sheet crystallinity in fibrous proteins by thermal analysis and infrared spectroscopy (DOI: 10.1021/ma0610109)” and “Degradation behavior of silk nanoparticles-enzyme responsiveness (DOI: 10.1021/acsbiomaterials.7b01021)”. The detailed descriptions of our physicochemical characterizations are listed as follows:

Firstly, we measured the UV absorption of silk fibroin aqueous solution. The UV spectra showed that the aqueous solutions of silk fibroin with different molecular weights did not differ significantly at the molecular level, as shown in Figure S3. An obvious ultraviolet characteristic absorption peak was observed at 275.5 nm, which was due to the presence of abundant aromatic amino acids in the molecular chain of silk fibroin [25].

Figure S3. UV spectra of silk fibroin aqueous solutions with different molecular weights. (A) RSF-0.5h; (B) RSF-1h; (C) RSF-2h.

Besides, the main functional group changes of silk fibroin obtained under different degumming time were investigated by FT-IR, and the influence of degumming time on the molecular conformation of silk fibroin was evaluated to a certain extent. The conformations such as β-sheet, α-helix, random coil and β-turn of silk fibroin molecular chain correspond to different absorption peak positions in the infrared absorption spectrum [s1], and the results are shown in Figure S4(A,C).

The infrared spectra of silk fibroin samples with different molecular weights are basically the same, and the characteristic absorption peaks around 3300 cm-1 and 3290 cm-1 are mainly attributed to the stretching vibration of -OH in the silk fibroin structure [s2].

Figure S4(A) shows that RSF-0.5h has strong characteristic absorption peaks at 1646 cm-1 (amide I), 1532 cm-1 (amide II) and 1236 cm-1 (amide III), amides I and II correspond to the α-helical conformation, and amide III corresponds to the random coil conformation. RSF-1h has strong characteristic absorption peaks at 1650 cm-1 (amide I), 1522 cm-1 (amide II) and 1236 cm-1 (amide III), amide I corresponds to the α-helix conformation, amide II corresponds to the β-sheet conformation, and amide III corresponds to the random coil conformation. RSF-2h has characteristic absorption peaks at 1649 cm-1 (amide I), 1521 cm-1 (amide II) and 1236 cm-1 (amide III), amide I corresponds to the α-helical conformation, amide II corresponds to the β-sheet conformation, and amide III corresponds to the random coil conformation. According to the protein secondary structure analysis of amides I, II and III, silk fibroin mainly exists in α-helix and random coil conformations in aqueous solution [s3].

Figure S4(C) shows that the lyophilized samples of silk fibroin hydrogels with three molecular weights have a strong characteristic absorption peak at around 1625 cm-1, 1520 cm-1 and 1230 cm-1, respectively. The peak shape is sharp, corresponding to the absorption peaks of silk fibroin amide I, amide II and amide III, respectively, showing a typical β-sheet conformation, in which amide III corresponds to a mixed conformation of β-sheet and random coil. According to the above protein secondary structure analysis of amides I, II and III, it can be seen that the silk fibroin in hydrogel mainly exists in the form of β-sheet, which mainly depends on the internal molecular chains of the silk fibroin. The crystalline regions formed by hydrogen bonds act as physical crosslinks, which slip during gel formation and freeze-drying and clump together to form a β-sheet conformation [s4].

Therefore, the silk fibroin in the aqueous solution belongs to the typical Silk I structure, and the extension of the degumming time may cause the structure to change from Silk I to Silk II; the silk fibroin in the hydrogel belongs to the typical Silk II structure, since the hydrogen bonds between molecular chains in Silk II structure are difficult to be broken, the performance of silk fibroin hydrogels is stable and insoluble in water [s5].

The position of the diffraction peak can reflect the related crystal structure of the silk fibroin material, and the intensity and width of the diffraction peak directly reflect the degree of crystallinity of the silk fibroin material [s6]. According to previous reports, in the XRD pattern of silk fibroin, 12.2°, 19.7°, 24.7°, 28.2°, 32.3°, 36.8°, and 40.1° are the main diffraction absorption peaks of Silk I structure (α-helix and random coil). 9.1°, 18.9°, 20.7°, 24.3°, 39.7° are the main diffraction absorption peaks of Silk II structure (β-sheet) [s7]. The crystal structures of silk fibroin samples of three molecular weights were analyzed by X-ray diffractometer.

Figure S4(B) shows that the three silk fibroin samples in the aqueous solution only have a “mound peak” with a large span in the range of 19.7°-24.7°, indicating that the silk fibroin in the aqueous solution lacks regular crystals. The region is dominated by Silk I structure (α-helix and random coil), that is, the “mound peak” peak near 22° can be identified as the non-crystalline region of silk fibroin. With the extension of degumming time, the peak-shaped structure changed from "short and wide" to "sharp and narrow", which was related to the change of the internal crystallinity of silk fibroin, and the destruction of the amorphous structure resulted in the formation of crystalline regions.

Figure S4(D) shows that the silk fibroin sample has a weak diffraction peak near 9.1°, and an obvious diffraction peak near 19.7°, indicating that there are two crystal structures of Silk I (α-helix and random coil) and Silk II (β-sheet) in silk fibroin hydrogel samples. It can be seen that some silk fibroin molecular conformation transitions from amorphous structure or Silk I (α-helix and random coil) structure to Silk II (β-sheet) crystalline structure, which is consistent with the results of infrared spectroscopy [s8].

Figure S4. Aggregated structure of silk fibroin. FT-IR (A) and XRD (B) spectra of silk fibroin aqueous solutions with different molecular weights. FT-IR (C) and XRD (D) spectra of silk fibro-in hydrogels with different molecular weights.

References

  • Hu, X.; Kaplan, D.L.; Cebe, P. Determining beta-sheet crystallinity in fibrous proteins by thermal analysis and infrared spectroscopy. Macromolecules 2006, 39, 6161-6170.
  • Wongpinyochit, T.; Johnston, B.F.; Seib, F.P. Degradation behavior of silk nanoparticles-enzyme responsiveness. ACS Biomater. Sci. Eng. 2018, 4, 942-951.
  1.  
  2.  
  3.  
  4.  
  5.  
  6.  
  7.  
  8.  
  9.  
  10.  
  11.  
  12.  
  13.  
  14.  
  15.  
  16.  
  17.  
  18.  
  19.  
  20.  
  21.  
  22.  
  23.  
  24.  
  25. Cheng, B.C.; Lei, Z.Y.; Wu, P.Y. Bio-derived crystalline silk nanosheets for versatile macroscopic assemblies. Nano Res. 2022, 10.1007/s12274-022-4124-x.
  • Dong, A.C.; Prestrelski, S.J.; Allison, S.D.; Carpenter, J.F. Infrared spectroscopic studies of lyophilization-and temperature-induced protein aggregation. Pharm. Sci. 1995, 84, 415-424.
  • Dong, Z.F.; Wang, Q.; Du, Y.M. Alginate/gelatin blend films and their properties for drug controlled release. Membrane Sci. 2006, 280, 37-44.
  • Panjapheree, K.; Kamonmattayakul, S.; Meesane, J. Biphasic scaffolds of silk fibroin film affixed to silk fibroin/chitosan sponge based on surgical design for cartilage defect in osteoarthritis. Design 2018, 141, 323-332.
  • Qi, Y.; Wang, H.; Wei, K.; Yang, Y.; Zheng, R.Y.; Kim, I.S.; Zhang, K.Q. A review of structure construction of silk fibroin biomaterials from single structures to multi-level structures. J. Mol. Sci. 2017, 18, 237-248.
  • Lopes, L.M.; Moraes, M.A.; Beppu, M.M. Phase diagram and estimation of flory-huggins parameter of interaction of silk fibroin/sodium alginate blends. Bioeng. Biotech. 2020, 8, 2296-2385.
  • Devi, D.; Sarma, N.S.; Talukdar, B.; Chetri, P.; Baruah, K.C.; Dass, N.N. Study of the structure of degummed Antheraea assamensis (muga) silk fibre. Text. Inst. 2011, 102, 527-533.
  • Ming, J.F.; Li, M.M.; Han, Y.H.; Chen, Y.; Li, H.; Zuo, B.Q.; Pan, F.K. Novel two-step method to form silk fibroin fibrous hydrogel. Sci. Eng. C-Mater. 2016, 59, 185-192.
  • Barud, H.G.O.; Barud, H.S.; Cavicchioli, M. Preparation and characterization of a bacterial cellulose/silk fibroin sponge scaffold for tissue regeneration. Polym. 2015, 128, 41-51.

Comment 2: BSA is a protein and so is silk fibroin. The authors should quantify the protein-protein interactions and how this is affecting or impacting the drug release from the hydrogels.

Our Reply: Thanks a lot for your insightful comment. BSA, as a drug model in this work, was encapsulated into silk fibroin hydrogel. The use of BSA for drug release investigations was a typical method, such as “A carbodiimide cross-linked silk fibroin/sodium alginate composite hydrogel with tunable properties for sustained drug delivery (DOI: 10.1002/mame.202100470)” and “Preparation and properties of O-chitosan quaternary ammonium salt/polyvinyl alcohol/graphene oxide dual self-healing hydrogel (DOI: 10.1016/j.carbpol.2022.119318)”. We are initially unaware of the protein-protein interactions between BSA and silk fibroin in this work. Actually, it exists and would influence the release behavior of BSA-loaded hydrogel, as well as its gelation and rheological properties. It's an interesting research topic. However, we are afraid that we have not enough data to well-explain this phenomenon. We can only provide our inferences below:

According to the rheological tests in Figure 3A and Figure S8 (Supplementary Materials), the storage modulus G' of blank silk fibroin hydrogel (without BSA-loading) was determined to be 0.77 kPa. After loading of BSA, the storage modulus G' increased, which meant a hydrogel with high strength was formed. Comparing to the blank silk fibroin hydrogel, the BSA-loading hydrogel with DLC of 20% possessed a G' of 7.88 kPa and the BSA-loading hydrogel with DLC of 40% possessed a G' of 9.28 kPa. Obviously, the more BSA was loaded, the higher hydrogel-strength exhibited. These results indicated the protein-protein interactions between BSA and silk fibroin in the gel-networks. However, we have not quantified them in our manuscript. This would be an interesting research topic in our further works.

References

  • Wang, Y.Y.; Niu, C.Q.; Shi, J.; Yu, W.L.; Zhu, C.H.; Zhang, Q.; Mizuno, M. A carbodiimide cross-linked silk fibroin/sodium alginate composite hydrogel with tunable properties for sustained drug delivery. Mater. Eng. 2021, 306, 2100470.
  • Cao, J.L.; He, G.H.; Ning, X.Q.; Chen, X.H.; Fan, L.H.; Yang, M.; Yin, Y.H.; Cai, W.Q. Preparation and properties of O-chitosan quaternary ammonium salt/polyvinyl alcohol/graphene oxide dual self-healing hydrogel. Polym. 2022, 287, 119318.

Figure 3. Rheological properties of silk fibroin hydrogels formed by silk fibroins with different Mw at 37 °C and concentrations of 30, 20, 15, 10 and 5 mg/mL. (A) RSF-1h; (B) RSF-0.5h.

Figure S8. Rheological properties of drug-containing hydrogels (A) DLC40 and (B) DLC20.

Reviewer 3 Report

The manuscript deals with the preparation nd characterization of silk fibroin hydrogels intended for protein drug delivery. It presents some severe concerns:

1) The novelty level of the manuscript is quite poor. The method of extraction of fibroin from silkworm cocoons is well established as well as the use of fibroin hydrogel for the delivery of protein drugs.

2) Many sentences are not written in correct English and all manuscript must be revised for language.

3) The style is imperfect since the organization of the work into the main manuscript and supplementary materials is not well performed. For instance, some experiments are reported in the main manuscript but the methods are missing in the main manuscript and reported in the supplementary materials without any referencing (such as for the degradation study or laser particle study). The organization of the work between manuscript and supplementary materials file must be enterely revised.

The scientific aim of the work must be highlighted and evidenced in the introduction. The introduction could be also shortened and be more focused.

The proposed spherical packing therory needs further experimental confirmations since samples are dried to perform SEM and structures from aqueous solution can easiliy collapse.

Scheme 1 seems a graphical abstract for the paper reporting the aim of recycling, the extraction method for silk and the gelation process. It could be not suitable to be inserted in the introduction.

The reuse of recycled silk is claimed but the silk used in the manuscript was extracted from raw silk textile yarns and not from recycled silk waste.

For micromorphology and degradation behaviour, it is not clear which RSF hydrogel (0.5 h, 1 h or 2h) was used.

It is not clear the formation of the fibroin hydrogel after exposure of the silk colloidal aqueous dispersions at 37 °C.

Why in Figure 5G and Figure 5H the experimental data at time 0 are not shown? In Lines 287-289 the effect of organic solvents is not clear.

Line 320-329 What the authors mean for drug loading? How the drug loading was calculated? How the maximum 20% of drug loading was determined for RSF10 and 70% for RSF15? How was selected the drug loading content of 40% and 20% for the release study? It is not undestandable how the different drug loading has a marked effect on the release (lines 333-335 and lines 358-359). Generally, the cumulant percentage release is dependent on the pharmaceutical dosage form and not dependent on the drug loading. Has the different viscosity of the two hydrogels an effect on BSA release?

Conclusions are a summary of the results and do not highlight the strenghts of the manuscript.

Author Response

Replies to your comments:

Firstly, we would like to express our great thanks for your constructive suggestions on our manuscript. After going through our paper carefully, we found that all the comments are helpful for us to revise our manuscript. Here, all scientific questions arisen from your comments are answered in detail one by one, hopefully the overall quality of this manuscript could reach the criteria for publishing in Molecules.

Response to the reviewer 3:

The manuscript deals with the preparation and characterization of silk fibroin hydrogels intended for protein drug delivery. It presents some severe concerns:

Comment 1: The novelty level of the manuscript is quite poor. The method of extraction of fibroin from silkworm cocoons is well established as well as the use of fibroin hydrogel for the delivery of protein drugs.

Our Reply: Thanks a lot for your thoughtful comment. The previously published research articles about this area are insightful and informative, and the method of extraction of fibroin from silkworm cocoons is well established as well as the use of fibroin hydrogel for the delivery of protein drugs. Compared with these published literatures, the main features and novelty of our work are elaborated below.

Firstly, in the published work, silk fibroin is mainly derived from silkworm cocoons, and there are relatively few studies on the extraction of silk fibroin from raw silk textile yarns. At the same time, our work has studied in detail the molecular weight distribution and micromorphology of silk fibroin in aqueous solution by regulating the degumming time of raw silk textile yarns, which has not been reported in previous published studies.

Secondly, in this work, the effects of concentration, pH and temperature on the gelation rate of silk fibroin solution were studied in detail, and a silk fibroin hydrogel with moderate molecular weight distribution and concentration, which could gel at physiological pH and temperature, was preferred.

Besides, we also comprehensively investigated the degradation behavior of silk fibroin hydrogel in different media. Moreover, the silk fibroin hydrogel prepared in this work has an excellent drug loading capacity, whose DLC can reach up to 70% to form an intact hydrogel without dehydration phenomenon.

Finally, we proposed and emphasized the idea of extracting silk fibroin from silk waste. We hope to develop a method of using silk fibroin hydrogel derived from raw silk textile yarns as protein drug carrier, and also hope that it can provide a preliminary reference for the recycled silk waste to be used in biomedical engineering.

Comment 2: Many sentences are not written in correct English and all manuscript must be revised for language.

Our Reply: Thanks for your suggestion. We regret there were problems with the English languages. We have thoroughly revised the manuscript and checked carefully to avoid typos. Furthermore, the manuscript has been carefully revised and re-polished by a professional language editing service (MDPI) to improve the grammar. The revised manuscript has been significantly polished. Therefore, the readers may understand our work more clearly. We hope it will meet a level suitable for reporting research in Molecules. The language polishing certificate is as follows. Finally, we sincerely appreciate your comments on English languages for giving us this precious opportunity to further ameliorate our manuscript.

Comment 3: The style is imperfect since the organization of the work into the main manuscript and supplementary materials is not well performed. For instance, some experiments are reported in the main manuscript but the methods are missing in the main manuscript and reported in the supplementary materials without any referencing (such as for the degradation study or laser particle study). The organization of the work between manuscript and supplementary materials file must be entirely revised.

Our Reply: Thank you for your time and offering us this valuable advice. We have reorganized the work between the main manuscript and supplementary material, and supplemented with recent relevant references for experimental methods.

The reorganization of the Materials and Methods section of the revised manuscript were labeled in red and presented below.

  1. Materials and Methods

3.1. Materials

Raw silk textile yarns from silkworm cocoons were purchased from Zhejiang Haiyan Jinyi Silk Spinning Co., Ltd.; Sodium carbonate, lithium bromide, sodium chloride, glutathione (GSH), N,N-dimethylformamide (DMF) and dimethyl sulfoxide (DMSO) were purchased from Aladdin Chemical Reagent Co., Ltd.; Phosphate-buffered saline (PBS) was purchased from Hangzhou Baisi Biotechnology Co., Ltd.; Anhydrous ethanol, hydrochloric acid and sodium hydroxide were purchased from Sinopharm Chemical Reagent Co., Ltd.; 3-(4,5-Dimethyl-2-thiazolyl)-2,5-diphenyltetrazolium bromide (MTT) was purchased from Sigma-Aldrich; Elastase (from porcine pancreas, 30 U/mg) was purchased from Beijing Jinming Biotechnology Co., Ltd.; Fluorescein isothiocyanate isomer I (FITC) was purchased from Innochem (Beijing, China); Bovine serum albumin (BSA) and dialysis bags (MWCO 3500 Da) were purchased from Shanghai Yuanye Bio-Technology Co., Ltd.; Liquid nitrogen was purchased from Yancheng Guangyuan Gas Co., Ltd. The chemical rea-gents were all analytical grade (AR), and the experimental water was ultrapure water/deionized water.

3.2. Testing and Characterization

The samples were treated by wet nitrification, and the lithium element in the sample solution was determined using an inductively coupled plasma emission spectrometer (ICP, OPTIMA 8000DV, PerkinElmer, USA). The silk fibroin aqueous solution was serially diluted with ultrapure water to a concentration range of 0.1-1.0 mg/mL. Based on the static light scattering (SLS) theory, the test temperature of the instrument was set to 25 °C, and the Malvern laser particle size analyzer (Zetasizer Nano ZS, Malvern, UK) was used to measure the light-scattering intensity of the sample solution to be tested relative to the standard with known Rayleigh ratio, and the molecular weight of silk fibroin was calculated by Rayleigh equation (1). Based on the dynamic light scattering (DLS) theory, the particle size distribution of the sample solution to be tested was measured by a Malvern laser particle size analyzer [41]. In addition, the gel time of the silk fibroin aqueous solution was observed and recorded at an ambient temperature of 4 °C.

KC/Rq = 1/M + 2A2C                            (1)

where C is the concentration, M is the molecular weight of the sample, A2 is the second virial coefficient, Rq is the Rayleigh ratio—the ratio of the scattered light of the sample to the incident light, and K is the optical constant.

The samples were tested for UV absorption using a full-wavelength UV spectrometer (UV-2450, Shimadzu, Japan), and the wavelength scanning range was set to 200-400 nm. The contents of C, N, and H in the samples were determined by elemental analyzer (Vario EL Cube, Elementar, Germany). A certain mass of freeze-dried samples was weighed and placed on the test bench of a Fourier-transform infrared spectrometer (FT-IR, NEXUS670, Nicolet, USA) for spectrum collection. The scanning range was 4000-400 cm-1, the number of scans was 64, and the resolution was 4 cm-1. The internal crystal structure of the samples was tested using an X-ray diffractometer (XRD, X’Pert3 Powder, PANalytical, Netherlands), where the diffraction angle range was 5-80°, the scanning speed was 2°/min, the current was 40 mA, the tube voltage was 40 kV, and CuKα rays were used. Under the protection of high-purity N2, the temperature was increased from room temperature to 800 °C at a heating rate of 10 °C/min, and the thermal stability of the samples was analyzed and determined using a TG-DSC synchronous thermal analyzer (STA 449C, NE-TZSCH, Germany).

3.3. Experimental Methods

3.3.1. Micromorphology Analysis

The silk fibroin aqueous solution was mixed with PBS buffer at a volume ratio of 1:1 and shaken, then placed in a 37 °C incubator to incubate until a hydrogel was formed. During this period, a small amount of solution was dropped on the activated silicon wafer every 12 hours, and left to stand overnight in a dry and clean area. After the water was fully evaporated and dried, the silicon wafer containing the sample was pasted on the electron microscope sample stage with conductive adhesive, and gold was sprayed with a spray current of 30 mA in a vacuum state for 2 min [42]. The microscopic morphology of the sample surface was observed under a Nova NanoSEM 450 scanning electron micro-scope with a shooting voltage of 15kV.

3.3.2. Rheological Analysis

The silk fibroin aqueous solution was diluted with ultrapure water to 30, 20, 15, 10, and 5 mg/mL, respectively, and incubated in a 37 °C incubator to form a hydrogel. The G' and G" of hydrogel samples with different contents of silk fibroin were tested in the "Time Mode" using a DHR-3 rheometer at a constant frequency of 10 rad/s, and the diameter of the test parallel plate was 40 mm [43]. The gap was set to 1 mm, and the temperature was set to 37 °C. In order to avoid the loss of water from the hydrogel, the samples should be tested in time after being quickly placed on the experimental bench of the instrument.

3.3.3. Cytotoxicity Assay

The cytotoxicity of silk fibroin hydrogels was evaluated by measuring the survival rate of human hepatoma cells (HepG2 cells) after co-culture in silk fibroin hydrogel leaching solution for a period of time by MTT assay. Under sterile conditions, we retrieved the 96-well plate in which the cells were grown adherently, and added 100 μL of silk fibroin hydrogel sample leaching solution and 100 μL of fresh DMEM medium containing 10% fetal bovine serum and 1% double antibody (100 U/mL penicillin-100 μg/mL streptomycin) to each well, and cultured the plate in a cell incubator at 37 °C with 5% CO2 [44]. After 48 h of incubation in the incubator, we added 20 μL of MTT solution (1 mg/mL) to each well. After a further 4 h of incubation, we carefully discarded the supernatant and added 150 μL of DMSO solution to each well, shook for 20 min, and measured the absorbance of each well with a microplate reader at a wavelength of 490 nm. In addition, the cell morphology (bright field) of the silk fibroin hydrogel sample leaching solution in the experimental group after co-culture with HepG2 cells for 48 h was photographed with an inverted microscope, and the morphology of the cells was compared with the control group to observe the morphology.

We calculated the relative cell viability according to Formula (2):

Relative cell viability (%) = As / A0                  (2)

where As is the absorbance value of the experimental group and A0 is the absorbance value of the control group.

In this work, an experimental group and a blank control group were set up. The experimental group contained the 24, 48 and 72 h leaching solutions of silk fibroin hydrogels (RSF-0.5h, RSF-1h and RSF-2h), and the leaching solutions were diluted 10 times; the blank control group was the medium containing 10% fetal bovine serum and 1% double antibody (100 U/mL penicillin-100 μg/mL streptomycin). The experimental group and blank control group were added to the 96-well plate inoculated with cells, and six parallel samples were set in each group. Finally, the 96-well plate was co-cultured in a cell incubator at 37 °C and 5% CO2 for 48 h.

3.3.4. Degradation in Different Media

The degradation behavior of two concentrations of silk fibroin hydrogels (RSF15 and RSF10) in different media (phosphate buffers with different pH values, hydrochloric acid and sodium hydroxide solutions with different concentrations, organic solvents, protease solutions, and NaCl solutions with different concentrations) were explored.

We prepared a number of dry and clean glass vials, numbered and weighed them, formed hydrogels in the glass vials, and weighed the total mass of the glass vials and the hydrogel after gelation. The fresh degradation solution was added to the glass vial containing the hydrogel at a ratio of 1:1 (v/v), and the degradation experiment was carried out in an incubator at 37 °C [45]. The samples were replaced with fresh degradation solution every day. The samples were taken out according to the set time, the degradation solution was discarded, and the total mass of the glass vial and the remaining hydrogel was weighed. The residual mass-retention rate of the hydrogel can be calculated according to Formula (3).

Remaining mass retention rate (%) = (Mi - M) / (M0 - M) × 100      (3)

where M0 is the initial total mass of the glass vial and hydrogel, g; M is the mass of the glass vial, g; Mi is the total mass of the glass vial and remaining hydrogel after i days, g.

3.3.5. Drug Loading and Release

The FITC-labeled BSA was used as a protein drug model, which was dissolved in PBS buffer and mixed by vortexing to obtain a drug solution with a specific concentration [46]. According to Formula 4, the drug loading content (DLC) was set to 10, 20, 30, 40, 50, 60, 70, 80 and 90%, and the concentration of the protein drug solution was calculated. The silk fibroin aqueous solution and the protein drug solution were mixed in equal volumes to obtain drug-containing silk fibroin solutions with final concentrations of 15 and 10 mg/mL, respectively. The drug-containing hydrogels can be obtained by incubation in a 37 °C incubator under dark conditions to promote drug encapsulation.

DLC represents the percentage of the total mass of the drug in the drug-containing hydrogel, and the specific calculation formula for this is as follows:

DLC (%) = cdVd / (cdVd + chVh) × 100                (4)

where cd is the concentration of the drug solution, mg/mL; Vd is the volume of the drug solution, mL; ch is the concentration of the pre-drug-loaded silk fibroin aqueous solution, mg/mL; Vh is the volume of the pre-drug-loaded silk fibroin aqueous solution, mL.

The PBS buffer was used as the drug-release solution, and each drug-containing hydrogel was tested in parallel in 3 groups. An equal volume of PBS buffer was added to the drug-containing hydrogel glass vial, and the drug was released in a 37 °C constant temperature incubator at a speed of 80 r/min. At the set time points (2, 4, 6, 8, 12, 24, 36, 48, 60, 72, 96, 120, 144, 168 h), we drew an appropriate amount of drug-release solution and immediately inject the same amount of fresh PBS buffer, keeping the volume of the release solution constant, and continuing to release the drug in the incubator. We dilute this sample of release solution 20 times with fresh PBS buffer, then diluted it in half and stepwise until it became a colorless and transparent liquid. We recorded the final dilution factor.

The fluorescence spectrum of the diluted drug-release solution was collected using a JASCO FP-6500 fluorescence spectrometer, and then the amount of the drug released at the corresponding time point was calculated according to the standard curve of fluorescently labeled BSA; then, the cumulative drug-release curve of the drug-containing hydrogel was drawn.

The final drug quality data calculated from the standard curve was the mean value of the three groups of parallel experiments, and the cumulative drug release rate of the drug-containing hydrogel was calculated according to Formula (5):

Cumulative release rate (%) = Mt / Md × 100              (5)

where Mt is the mass of the drug released at time t, mg; Md is the initial mass of the loaded drug, mg.

References

  1.  
  2.  
  3.  
  4.  
  5.  
  6.  
  7.  
  8.  
  9.  
  10.  
  11.  
  12.  
  13.  
  14.  
  15.  
  16.  
  17.  
  18.  
  19.  
  20.  
  21.  
  22.  
  23.  
  24.  
  25.  
  26.  
  27.  
  28.  
  29.  
  30.  
  31.  
  32.  
  33.  
  34.  
  35.  
  36.  
  37.  
  38.  
  39.  
  40.  
  41. Liu, J.W.; Ding, Z.Z.; Lu, G.Z.; Wang, J.G.; Wang, L.; Lu, Q. Amorphous silk fibroin nanofiber hydrogels with enhanced mechanical properties. Biosci. 2019, 19, 1900326.
  42. Cui, Y.J.; Zhang, F.; Chen, G.; Yao, L.; Zhang, N.; Liu, Z.Y.; Li, Q.S.; Zhang, F.L.; Cui, Z.Q.; Zhang, K.Q.; Li, P.; Cheng, Y.; Zhang, S.M.; Chen, X.D. A stretchable and transparent electrode based on PEGylated silk fibroin for in vivo dual-modal neural-vascular activity probing. Mater. 2021, 33, 2100221.
  43. Wang, X.Q.; Partlow, B.; Liu, J.; Zheng, Z.Z.; Su, B.; Wang, Y.S.; Kaplan, D.L. Injectable silk-polyethylene glycol hydrogels. Acta Biomater. 2015, 12, 51-61.
  44. Xu, Z.P.; Chen, T.Y.; Zhang, K.Q.; Meng, K.; Zhao, H.J. Silk fibroin/chitosan hydrogel with antibacterial, hemostatic and sustained drug-release activities. Int. 2021, 70, 1741-1751.
  45. Zhang, W.; Chen, J.L.; Qu, M.L.; Backman, L.J.; Zhang, A.N.; Liu, H.Y.; Zhang, X.P.; Zhou, Q.J.; Danielson, P. Sustained release of TPCA-1 from silk fibroin hydrogels preserves keratocyte phenotype and promotes corneal regeneration by inhibiting interleukin-1β signaling. Healthc. Mater. 2020, 9, 2000591.
  46. Wang, Y.Y.; Niu, C.Q.; Shi, J.; Yu, W.L.; Zhu, C.H.; Zhang, Q.; Mizuno, M. A carbodiimide cross-linked silk fibroin/sodium alginate composite hydrogel with tunable properties for sustained drug delivery. Mater. Eng. 2021, 306, 2100470.

Comment 4: The scientific aim of the work must be highlighted and evidenced in the introduction. The introduction could be also shortened and be more focused.

Our Reply: Thanks for your constructive suggestion. We agree that the introduction section should be shortened and more focused. We have rewritten the introduction section in the revised manuscript and it’s presented below. In this section, many redundant and basic descriptions have been removed or concentrated. We have also paid more attention on the main theme of this manuscript. We have proposed and emphasized the aims of this work in the revised introduction section. Silk fibroin is mainly derived from silkworm cocoons in the published works. To the best of our knowledge, studies on the extraction of silk fibroin from raw silk textile yarns to prepare hydrogels as protein drug carriers are rare. We developed a method to use silk fibroin hydrogel derived from raw silk textile yarns as drug carrier in this manuscript, and hope that it could be a potential utility for the recycled silk waste to be used in biomedical area. In addition, the gelation properties of silk fibroin hydrogels under different conditions were investigated in detail. We found that the concentrations of RSF, the environmental temperatures and the pH values would all influence the gelation behaviors of silk fibroin. On the other hand, the degradation experiments of silk fibroin hydrogel were evaluated in a various of media, which presented its interesting degradation behaviors under different conditions.

Comment 5: The proposed spherical packing theory needs further experimental confirmations since samples are dried to perform SEM and structures from aqueous solution can easily collapse.

Our Reply: Thanks for your constructive suggestion. According to the SEM observation, it can only provide preliminary evidence that silk fibroin exists in the form of "spherical" aggregate particles after drying. Indeed, further characterizations are needed to clarify the "spherical packing" theory. Therefore, after comprehensive consideration, we have removed all the relevant descriptions about the "spherical packing" theory in the revised manuscript.

Comment 6: Scheme 1 seems a graphical abstract for the paper reporting the aim of recycling, the extraction method for silk and the gelation process. It could be not suitable to be inserted in the introduction.

Our Reply: Thanks for your constructive suggestion. After careful consideration, we thought it would be more appropriate to place Scheme 1 in the "2. Results and Discussion" section. Therefore, we moved Scheme 1 to the "2.1. Preparation and Characterization of Silk Fibroin" section. The new location for Scheme 1 is pasted below.

2.1. Preparation and Characterization of Silk Fibroin

Degummed silk with different molecular weights can be obtained by heating and boiling for different time (0.5, 1 and 2 h) under the same power. The silk fibroin extracted under degumming times of 0.5, 1 and 2 h were named as RSF-0.5h, RSF-1h and RSF-2h, respectively. The prepared degummed silk was dissolved in 9.3 mol/L lithium bromide (LiBr) solution to obtain silk fibroin aqueous solution with three molecular weights. The preparation process of silk fibroin aqueous solution is shown in Scheme 1B [23].

Scheme 1. (A) A promising strategy for using recycled silk waste to take the place of silk cocoons from natural sources to prepare silk-based hydrogels as protein/peptide carriers. (B) Procedure to obtain silk fibroin aqueous solution. The raw silk textile yarns were degummed by the secondary boiling method with sodium carbonate solution. The obtained degummed silk was dissolved in 9.3 mol/L lithium bromide solution, followed by a series of processes such as dialysis, filtration and concentration determination to prepare a silk fibroin aqueous solution with a concentration of 30 mg/mL. (C) The sol–gel transformation process at pH 7.4 under the temperature of 37 °C.

Comment 7: The reuse of recycled silk is claimed but the silk used in the manuscript was extracted from raw silk textile yarns and not from recycled silk waste.

Our Reply: Thanks a lot for your constructive comment. Indeed, in order to prepare silk fibroin hydrogel as a protein drug carrier, we proposed an idea to extract natural silk fibroin from the recycled silk waste in the introduction section, which might be a potential utility in the industrial area. However, in the practical investigations and characterizations, we found that the silk fibroin from recycled silk waste might contain various pigments and additives. And some of them might be the mixtures of natural silk and other synthetic polymers, which would bring about unfavorable influences on the physical-chemical properties of silk fibroin hydrogel, including the gelation behaviors and the biocompatibilities, etc.. On the other hand, the separation and purification procedures of silk fibroin from recycled silk waste could be a harsh task, which needed further developments. Therefore, in this work, the pure silk fibroin was extracted from the raw silk textile yarns instead of the recycled silk waste, considering some of the commercial silk products are made of raw silk textile yarns.

By the way, we are currently optimizing our purification methods and establishing an economic way to obtain clean silk from recycled silk waste products.

Comment 8: For micromorphology and degradation behavior, it is not clear which RSF hydrogel (0.5 h, 1 h or 2h) was used.

Our Reply: Thanks for your thoughtful suggestion. We are aware of this issue. We used silk fibroin (RSF-1h) degummed for 1h in the two parts of "Micromorphology Analysis" and "Degradation Behavior Evaluation". We have added explanatory words where appropriate in the revised manuscript, and labeled in red.

2.2. Micromorphology Analysis of Silk Fibroin Sol-Gel Transformations

The sol–gel transformations process of silk fibroin aqueous solution (RSF-1h) with concentrations of 15 mg/mL (RSF15) and 10 mg/mL (RSF10) was observed by SEM, and the results are shown in Figure 2. The silk fibroin was dispersed in the aqueous solution in the form of "spherical" aggregate particles. With the extension of time, the silk fibroin in the aqueous solution will spontaneously aggregate to form larger "spherical" particles. The measured particle sizes of RSF15 at 0, 12 and 24 h were 47.41±1.72, 107.48±2.89 and 155.48±3.91 nm, and the particle sizes of RSF10 at 0, 12 and 24 h were 38.07±4.17, 85.41± 4.78 and 145.56 ± 2.18 nm, respectively. Obviously, the particle size will increase with the extension of time. Before gelation, the "pellets" were continuously clustered together into "big balls" and then accumulate with each other to form aggregates. The aggregates were interconnected to form a porous network with a certain spatial conformation, thereby forming a hydrogel.

Figure 2. Morphologies of silk fibroin sol-gel transformation process. Scale bar is 500 nm.

2.5. Degradation Behavior Evaluation of Silk Fibroin Hydrogels

The degradation behavior of hydrogels formed from silk fibroin (RSF-1h) obtained by degumming for 1 h was investigated in this work. Figure 6(A, B) showed that the silk fibroin hydrogel had no obvious degradation in PB buffers with different pH values, but displayed a slight swelling phenomenon. Only RSF10 showed degradation in PB buffers with pH 6.0 (Figure 6B), at a rate of about 22% on the 22nd day. This may be due to the partial acidolysis of the protein in the weak acid environment. The silk fibroin hydrogel had a three-dimensional porous network structure with a large specific surface area [34] which can absorb and lock part of the water; in addition, some salt ions entered the hydrogel, resulting in dense pores, limiting the water absorption capacity of the hydrogel, and reaching a swelling balance [24].

Comment 9: It is not clear the formation of the fibroin hydrogel after exposure of the silk colloidal aqueous dispersions at 37 °C.

Our Reply: Thanks for your comment. The formation of hydrogel from silk fibroin solution at 37 ℃ is a slow physical spontaneous crosslinking process. The recent published reviews about this area are insightful and informative, such as the paper “Functional silk fibroin hydrogels: preparation, properties, and applications (DOI: 10.1039/d0tb02099k)”. Referring to this literature, it is clear from our work that, in the regeneration solution, silk fibroin mainly presents in the form of Silk I structure. It can induce the silk fibroin structure to acquire a lower-energy β-sheet conformation at 37 °C, and then form a hydrogel under self-aggregation.

Comment 10: Why in Figure 5G and Figure 5H the experimental data at time 0 are not shown? In Lines 287-289 the effect of organic solvents is not clear.

Our Reply: Thanks for your reminding. We carefully rechecked Figure 5 and found that the experimental data at time 0 were not shown in partial graphs. To this end, we supplemented the corresponding experimental data at time 0 in the partial graphs, and redrew Figure 5. The redrawn Figure 5 is pasted below and presented in the revised manuscript.

Furthermore, in this work, we described the observed phenomena after immersion of silk fibroin hydrogels in organic solvents for 22 days. We have observed the phenomena, but the effect of organic solvents on silk fibroin hydrogels cannot be explained based on the existing data, which might be investigated in the future work to give a reasonable explanation.

Figure 5. Degradation behaviors of silk fibroin hydrogels in different media. (A, C, E, G, I, K) were the hydrogels formed by the silk fibroin solutions with the final concentration of 15 mg/mL; (B, D, F, H, J, L) were the hydrogels formed by the silk fibroin solutions with the final concentration of 10 mg/mL. The freshly prepared silk fibroin aqueous solution was diluted to 30 and 20 mg/mL with ultrapure water. Then, the standby silk fibroin aqueous solution was mixed with PBS buffer at a volume ratio of 1:1 and shaken to obtain silk fibroin solutions of 15 and 10 mg/mL, denoted as RSF15 and RSF10, respectively, and placed in a 37 °C incubator for static incubation to form hydrogels. These were used to study the degradation behavior in different media.

Comment 11: Line 320-329 What the authors mean for drug loading? How the drug loading was calculated? How the maximum 20% of drug loading was determined for RSF10 and 70% for RSF15? How was selected the drug loading content of 40% and 20% for the release study? It is not understandable how the different drug loading has a marked effect on the release (lines 333-335 and lines 358-359). Generally, the cumulant percentage release is dependent on the pharmaceutical dosage form and not dependent on the drug loading. Has the different viscosity of the two hydrogels an effect on BSA release?

Our Reply: Thanks for your comments. The detailed answers to this question are presented as follows.

Drug loading content (DLC) represents the percentage of the total mass of the drug in the drug-containing hydrogel, and the specific calculation formula is as follows:

DLC (%) = cdVd / (cdVd + chVh) × 100

where cd is the concentration of the drug solution, mg/mL; Vd is the volume of the drug solution, mL; ch is the concentration of the pre-drug-loaded silk fibroin aqueous solution, mg/mL; Vh is the volume of the pre-drug-loaded silk fibroin aqueous solution, mL.

The photographs of RSF10 FITC-BSA-containing hydrogels with different DLCs are shown in Figure S6. By the increasing of FITC-BSA loading content up to 30%, the hydrogels presented an undesirable dehydration phenomenon. While the DLC of RSF10 was kept to 20% or below, an intact FITC-BSA-loaded silk fibroin hydrogel could be formed. Therefore, the maximum DLC of RSF10 to form an intact hydrogel was 20%, and all fed FITC-BSA drug was encapsulated into the hydrogel in this process with a drug loading efficiency of 100%. Besides, the photographs of RSF15 FITC-BSA-containing hydrogels with different DLCs are shown in Figure S7. According to the observations, the maximum DLC of RSF15 to form an intact hydrogel without dehydration was determined to be 70%. The drug loading efficiency in this hydrogel was also 100%.

Figure S6. Photographs of RSF10 drug-containing hydrogels with different drug loadings.

Figure S7. Photographs of RSF15 drug-containing hydrogels with different drug loadings.

To avoid the waste of silk fibroin, a high DLC to form intact drug-containing hydrogel is beneficial. However, it was observed that higher DLC led to higher gel-strength, but exhibiting dehydration phenomenon, as shown in Figure S6, Figure S7 and Figure S8. In order to obtain appropriate FITC-BSA-loaded silk fibroin hydrogels with moderate DLCs and adequate gel-strengths as the candidates for the evaluation of drug release profiles, the final DLCs of BSA-loaded hydrogel in this experiment were set as 20% and 40%. In detail, the silk fibroin solution with an initial concentration of 30 mg/mL and the FITC-BSA solution were selected for equal volume mixing to prepare FITC-BSA-loading hydrogels with the DLCs of 40% and 20% for drug release experiments. After drug encapsulation, the hydrogels with DLCs of 40% and 20% are tagged as DLC40 and DLC20, respectively. And the final RSF concentrations in the two hydrogels were 15 mg/mL. Their drug release profiles were shown in Figure 7 of the revised manuscript.

Figure 7. Cumulative drug release curve of drug-containing hydrogels.

The statement “but the cumulative drug-release rate of DLC40 is higher than that of DLC20 at the same point in time. This is because the silk fibroin hydrogel has a porous structure, and its internal drug concentration is high and easily diffuses and releases outward.” in Sub-section 2.6 of the original manuscript is not understandable and has been removed. With regards to this, we rephrased this sentence as “and the cumulative drug-release rate of DLC40 was higher than that of DLC20 at the same point in time. This is because DLC40 encapsulated a higher concentration of BSA, which will lead to more diffusion of BSA molecules into the release solution.” in Sub-section 2.6 of the revised manuscript.

Figure S8 shows the variation in the storage modulus G' and the loss modulus G" of DLC40 and DLC20 after gelation. It was observed that the G" values of DLC40 are close to that of DLC20. Therefore, in this work, we considered that the viscosity has little effect on the release profile of BSA from silk fibroin hydrogel.

Figure S8. Rheological properties of drug-containing hydrogels (A) DLC40 and (B) DLC20.

Comment 12: Conclusions are a summary of the results and do not highlight the strengths of the manuscript.

Our Reply: Thanks for your thoughtful comment. According to your comments, we re-summarize the work of the full text and have revised the Conclusions section in the revised manuscript as presented below. In this section, many unnecessary descriptions have been removed.

Conclusions Section of the revised manuscript:

In conclusion, a complete and stable silk fibroin extraction and aqueous solution preparation process was established by using raw silk textile yarns as stock materials. Silk fibroins with different molecular weights can be obtained by controlling the degumming time. The silk fibroin was dispersed in the aqueous solution as "spherical" aggregate particles, and smaller particles continuously accumulated into larger particles. Finally, a silk fibroin hydrogel network was formed. The secondary structures of silk fibroin aqueous solution and hydrogel lyophilized samples were different. The former was dominated by Silk I structure, while the latter was dominated by Silk II structure, and the crystallinity of the hydrogel samples was higher. When the silk fibroin hydrogel concentration increased, its storage modulus increased significantly. By studying the degradation behavior of silk fibroin hydrogel in various media, it had been verified that it had excellent stability, which can further broaden its potential applications. Moreover, the prepared silk fibroin hydrogel had good biocompatibility and excellent drug loading capacity, and the cumulative drug release reached 80% within 12 h after loading BSA, indicating the potential utility of this hydrogel for the delivery of protein drugs.

Round 2

Reviewer 1 Report

The manuscript could be published in the revised form.

Reviewer 3 Report

The manuscript is suitable for publication